# Molecular profiles, sources and lineage restrictions of stem cells in an annelid regeneration model

Alexander W. Stockinger [1,2,3,4,5,12], Leonie Adelmann [1,2,3,4,5,12], Martin Fahrenberger [1,3,4,6,7], Christine Ruta [8], B. Duygu Özpolat [9,10], Nadja Milivojev [1,2,3,4,5], Guillaume Balavoine [9,11] ✉ & Florian Raible [1,2,3] ✉

Regeneration of missing body parts can be observed in diverse animal phyla, but it remains unclear to which extent these capacities rely on shared or divergent principles. Research into this question requires detailed knowledge about the involved molecular and cellular principles in suitable reference models. By combining single-cell RNA sequencing and mosaic transgenesis in the marine annelid *Platynereis dumerilii,* we map cellular profiles and lineage restrictions during posterior regeneration. Our data reveal cell-type specific injury responses, re-expression of positional identity factors, and the re-emergence of stem cell signatures in multiple cell populations. Epidermis and mesodermal coelomic tissue produce distinct putative posterior stem cells (PSCs) in the emerging blastema. A novel mosaic transgenesis strategy reveals both developmental compartments and lineage restrictions during regenerative growth. Our work supports the notion that posterior regeneration involves dedifferentiation, and reveals molecular and mechanistic parallels between annelid and vertebrate regeneration.

The ability of some animals to regenerate missing body parts is a fascinating phenomenon. Whereas the regenerative ability of mammals is generally limited to individual cell types or a few specific organs, other animals are capable of rebuilding their entire body from mere fragments of tissue. Complex tissue regeneration can be observed in almost all clades of bilaterian life and might therefore reflect an ancient capacity[1,2]. This regenerative process usually involves the formation of an epimorphic, proliferative cell mass called blastema[3]. The cellular sources and molecular properties of blastema cells, however, differ

between available model systems. For example, whereas some invertebrates like the planarian *Schmidtea mediterranea* form their blastemas from totipotent stem cells[4], other regenerative models employ less potent stem cells, such as uni- or oligopotent progenitors, and make use of dedifferentiation, transdifferentiation or a combination of these processes[3,5–7]. While molecular similarities between these regenerative strategies can be observed across phyla, we still lack comprehensive data on representative species to uncover whether or not these similarities indicate true homologies[3]. Defining molecular

¹Max Perutz Labs, Vienna Biocenter Campus (VBC), Vienna, Austria. ²University of Vienna, Center for Molecular Biology, Department of Genetics and Microbiology, Vienna, Austria. ³Research Platform Single-Cell Regulation of Stem Cells (SinCeReSt), University of Vienna, Vienna, Austria. ⁴Vienna Biocenter PhD Program, a Doctoral School of the University of Vienna and the Medical University of Vienna, Vienna, Austria. ⁵PhD Programme Stem Cells, Tissues, Organoids – Dissecting Regulators of Potency and Pattern Formation (SCORPION), University of Vienna, Vienna, Austria. ⁶Center for Integrative Bioinformatics Vienna (CIBIV), University of Vienna and Medical University of Vienna, Vienna, Austria. ⁷Medical University of Vienna, Max Perutz Labs, Vienna, Austria. ⁸Institute of Biology, Federal University of Rio de Janeiro, Rio de Janeiro, Brazil. ⁹Université de Paris Cité, CNRS, Institut Jacques Monod, Paris, France. ¹⁰Present address: Department of Biology, Washington University in Saint Louis, St. Louis, MO, USA. ¹¹Present address: Institute of Neuroscience, CNRS, Université Paris-Saclay, Saclay, France. ¹²These authors contributed equally: Alexander W. Stockinger, Leonie Adelmann. ✉e-mail: guillaume.balavoine@cnrs.fr; florian.raible@univie.ac.at

signatures of blastema formation in accessible model systems and regeneration paradigms will be a key requirement for such comparisons.

Annelids show particular promise as models to inform mechanisms and pathways of blastema-based regeneration. Many species within this clade exhibit the ability to regenerate large parts of their primary body axis. Annelids are phyletically well-positioned for long-range comparisons with other lophotrochozoan clades, such as planarians, as well as deuterostomes (which include vertebrates). Likewise, comparisons between different annelids provides an avenue to assess modulation of regenerative capacities within the clade[8–12]. This makes annelids ideal models to assess both commonalities and differences in regenerative processes.

The nereidid worm *Platynereis dumerilii* has a long history as a model system for regeneration and regenerative growth[8,9]. This bristleworm can be continuously bred in the laboratory, and offers a variety of molecular and genetic tools for analysis, including transcriptomic profiling, multiplexed detection of RNAs in fixed specimens, and transgenic manipulation[9,13–17].

During normal development, *Platynereis* grows by continuous segment addition, which involves a dedicated ring-shaped "segment addition zone" (SAZ) located between the posterior-most segment and the post-segmental pygidium[18]. Molecularly, the SAZ harbors putative posterior stem cells (PSCs) that express members of the germline multipotency program (GMP)[19], such as *piwi*, *vasa* and *nanos*.

Upon amputation across the primary body axis, *Platynereis* re-establishes a functional SAZ which then produces the missing posterior segments. Morphologically, this process has been well characterized (reviewed in refs. 8,10,20): in a first, rapid response to injury, the gut seals the wound. In a second step, epidermal cells cover the injury under a wound epithelium, followed by the formation of a largely undifferentiated blastemal cell mass, from which the new SAZ emerges. After this point, new segments are added and the animals grow faster than during regular development[10,21].

The source of stem cells during regeneration has been of long-standing interest. Tissue-residual stem cells, as well as the de- and trans-differentiation of somatic cells have been observed as sources of animal blastemas (reviewed in refs. 5,6,22). While adult stem cells as a source have been described in some invertebrates, such as planarians (see above), most data within annelids point towards de-differentiation as a likely source of stem cells during regeneration (reviewed in refs. 8,20), with few exceptions found among the class *Clitellata* (reviewed in ref. 10).

In *Platynereis*, early cytological evidence already suggested the amputation-induced emergence of new stem cells in differentiated tissues such as the wound-adjacent epidermis. This process has classically been referred to as "re-embryonalisation"[23]. Re-amputation, along different planes of posterior regenerates that had been labeled using EdU (5'-ethynyl-2'-deoxyuridine) incorporation, suggests that resident proliferating cells do not contribute disproportionately to the regenerating SAZ. It has therefore been suggested that the *Platynereis* SAZ regenerates from wound-adjacent, presumably differentiated cells, which activate GMP gene expression and re-enter the cell cycle after injury[21].

24 h post amputation (hpa), the wound is covered with an epithelium. This stage can be reached even when proliferation is inhibited[21]. At 24 hpa, several genes usually found in the SAZ of uninjured animals, such as the ectodermal PSC marker *hox3* and the GMP members *piwi* and *myc*, are expressed de novo in and near the wound.

After 48 hpa, injured worms have formed a blastema, at which point proliferation increases markedly[21]. Bulk transcriptomic profiling and an analysis of epigenetic factor expression during regeneration support the idea that these steps are accompanied by chromatin remodeling[24,25]. While other cell sources, such as quiescent and currently undiscovered residual stem cells can not be fully excluded, these findings support the notion of differentiated cells providing the source for a regenerating SAZ through dedifferentiation.

Currently available data lack the cellular resolution to identify tissue- and cell type specific properties of regeneration in *Platynereis*, including the transcriptional profile of cells responding to injury with chromatin remodeling and GMP gene expression as outlined above. Additionally, no information regarding the differentiation potential / lineage restriction of *Platynereis* PSCs is currently available, further complicating comparative analyses. To gain deeper and unbiased insight into this process and enable cross-species comparisons of blastema based regeneration, molecular profiling at cellular resolution and clonal information of lineage restriction are required.

Here, we follow a dual approach of single-cell RNA sequencing and transgenesis experiments throughout posterior regeneration to address these challenges. By sampling single-cell transcriptomes at multiple regenerative stages and comparing them to wound-adjacent tissue right after injury, we are able to derive a comprehensive, time-resolved map of cellular profiles over the regenerative process. We detect cell-type specific injury responses and re-expression of positional identity factors. We also uncover that multiple wound-adjacent cell populations start expressing stem cell related genes and enter the cell cycle upon injury, consistent with the notion that these cells undergo dedifferentiation.

Investigating signature genes for two of these populations, we identify the epidermis and mesodermal coelomic tissue as two likely source tissues that produce distinct PSCs in the segment addition zone. Capitalizing on a novel mosaic transgenesis strategy, we are able to identify both developmental compartments and restrictions in cell lineages throughout posterior growth and regeneration. We demonstrate that the SAZ of *Platynereis dumerilii* harbors separate pools of lineage-restricted PSCs and that these pools are regenerated from cells originating from distinct embryonic germ layers. Our combined datasets provide a detailed view of the sources, molecular signatures and differentiation potential of major cell types in the blastema, and reveal molecular and mechanistic similarities between annelid and vertebrate regeneration.

## Results

### A dynamic transcriptional landscape of posterior regeneration

To establish a first unbiased, in-depth analysis of the transcriptomic landscape of individual cell populations during posterior regeneration in annelids, we devised a suitable sampling scheme. For this, we induced posterior regeneration by removing approximately 1/3 of the animals' posterior tissue, including the SAZ and its rapidly proliferating progeny, amputating between segments 30 and 31. For sampling, we then isolated the posteriormost segment along with any newly regenerated tissue at distinct time points after amputation (Supplementary Fig. 1a).

We reasoned that inclusion of the last non-amputated segment in these analyses would not only provide us with data on differentiated cell types, but also allow us to detect any molecular signatures associated with the response of this segment to the adjacent wound. To assess whether this sampling scheme captured relevant molecular events in the early phases of blastema formation, we first performed a bulk RNA sequencing experiment, in which the total mRNA of each sampling time point was sequenced from biological triplicates. By using an unbiased gene-clustering approach, we determined seven major categories of gene expression dynamics over the first three days of regeneration, including four categories in which gene expression increased after amputation, with differences in the point of onset and kinetics (Supplementary Fig. 1b). Genes in these categories include known markers for stem cells and the SAZ, as well as proliferation-related transcripts (Supplementary Fig. 1b). These findings are consistent with previous observations[21,24,25] and confirmed that our sampling strategy could be used to capture relevant molecular processes.

Based on these results, we devised a similar sampling scheme to build a comprehensive single cell atlas of posterior regeneration (Fig. 1a). We obtained single cells from multiple dissociated samples, representing five distinct stages of regeneration. These spanned from freshly amputated individuals (0 hpa, equivalent to uninjured trunk segments, but not the posterior-most tissues such as the SAZ and its immediate progeny) to 72 hpa, corresponding to the onset of rapid proliferation in the regenerated SAZ[21], increasing the temporal resolution in early regenerative stages by adding a 12 hpa timepoint (Fig. 1a). After removing outlier and low-quality cells, we obtained a total of 80,298 transcriptomes of individual cells, sampled in two independent biological replicates of 4 and 5 timepoints, respectively (Supplementary Data 1). Even though the sampling timepoints of this single cell experiment slightly differed from those sampled in bulk (see above), we compared the two datasets using a correlation analysis. Despite the use of different sampling, sequencing and processing techniques, all replicates correspond most strongly with those in the respective other dataset sampled at the closest time point (Supplementary Fig. 1c).

Single-cell data comprising multiple replicates or biological samples might suffer from batch effects, where technical differences between sampling rounds could overshadow biologically meaningful differences between samples or cell types[26]. To counter this effect and minimize technical variations, we took advantage of the recent establishment of a combined cell fixation and storage protocol (ACetic-MEthanol/ACME) that is compatible with single-cell sequencing[27]. The adaptation of this protocol for our *Platynereis* regenerate paradigm allowed us to sort, process and sequence cells from all sampled stages in parallel. We subsequently used standard single-cell RNA sequencing (scRNA-seq) analysis methods to process the joined data-set (see Methods).

Unbiased clustering of the cells resulted in 38 transcriptionally distinct clusters. The comparison between biological replicates and timepoints did not suggest any batch effect affecting cluster formation (Supplementary Data 1, Supplementary Fig. 2a–e). The resulting clusters, as illustrated on a uniform manifold approximation and projection (UMAP) visualization[28] (Fig. 1b), correspond to cell populations of similar transcriptomic profiles. Algorithmic prediction[29] identifies one cluster (cluster 26) as the possible product of doublet formation, so this cluster was not investigated further (Supplementary Data 1). We annotated these populations based on the identities of cluster-specific marker genes, and their expression levels of known annelid cell-type markers. In total, we annotated 35 of the clusters, either as known cell populations, or based on their most diagnostic marker gene (Fig. 1b, see details in Supplementary Data 2 and 3).

As each sampled tissue contains the segment adjacent to the injury site, we were able to identify a variety of cell types in our dataset. For example, an investigation of genes previously used for assigning different *Platynereis* muscle cell types[30], allowed us to distinguish several populations of smooth (clusters 3, 6, 8, 12 and 14) and striated (clusters 2, 10, and 17) muscle. Even less abundant cell types, such as chaetal sac cells (cluster 24) which form the bristle worm's chitinous bristles[31,32] and extracellular globin-secreting cells (cluster 15)[33], were identified as distinct populations. This shows that our approach yielded a high-quality cell atlas containing biologically meaningful clusters of cell populations and with sufficient sensitivity to resolve rare and poorly understood cell types.

**Molecular repatterning and emerging stem cell-like properties**
As outlined above, deconvolving the dynamic injury response to individual cell populations in an annelid is expected to advance our understanding of regeneration in an evolutionary context. By capitalizing on the temporal information embedded in each transcriptome of our dataset (Fig. 1c), we were able to perform comparisons of gene expression within cell populations across time.

A common challenge in complex tissue regeneration is the re-establishment of appropriate positional information, such as the position along the antero-posterior axis. To test whether our dataset could be used to identify the individual cell types involved in repatterning, we analyzed the expression of several transcription factors involved in posterior identity. Bulk RNA sequencing of posterior regeneration and unbiased clustering of genes with similar expression dynamics using mfuzz (Supplementary Fig. 1b) revealed the presence of genes encoding posteriorly expressed transcription factors such as *caudal* (*cdx*), *distalless* (*dlx*) and *foxA*, in gene sets upregulated after injury. This is consistent with previous suggestions that early steps in annelid regeneration include a morphallactic adjustment of positional values[34].

Using the single cell atlas, we were able to add cellular resolution to this process. For example, cells of midgut identity (cluster 16) are only found in the freshly amputated sample (0 h post amputation, hpa), subsequently yielding to a population (cluster 4) demarcated by *foxA* and *cdx* as hindgut after injury (Fig. 1d, e). This morphallactic process of gut posteriorization indicated by *foxA* has previously been proposed in *Platynereis*[35], demonstrating the validity of our in silico approach. In addition, a subset of neuronal populations (clusters 11, 20) expresses *cdx* and *foxA* shortly after injury (Fig. 1d, e), while two other populations (clusters 0, epithelium; cluster 9, *gcm*+ neurons) started to express *dlx* (Supplementary Fig. 3a). Similarly, we observed a molecular shift in presumptive smooth muscle cells from cluster 14 (pre-injury) to clusters 8 (post-injury), involving genes like *thrombospondin*, *rho kinase*[36] and *octopamine receptor 2*, which play a role in muscle attachment, function and regeneration in other species[37–40] (Supplementary Data 3).

As these data supported our approach to reconstructing temporal dynamics, we next investigated the expression of stem cell related genes after posterior injury. We reasoned that if stem cells are, at least in part, regenerated by dedifferentiation or activation of wound-adjacent cells, we should detect cell populations that are already present at 0 hpa, but start to express stem cell and proliferation-related markers only after injury.

To assess this point, we first investigated the expression of the homeobox gene *hox3*, whose transcripts are rapidly upregulated in posterior regeneration of *Platynereis dumerilii*[41] and mostly restricted to a population of PSCs that are generally referred to as ectodermal PSCs in accordance with their presumed developmental origin[18,21]. Whereas homeostatic trunk cells (0 hpa) are almost entirely devoid of *hox3* expression, we could detect a strong and mostly cluster-specific upregulation of this gene in post-injury time points of cluster 0 (Fig. 1f). Likewise, we find that this cluster expresses *Platynereis piwi* (Fig. 1g), a key member of the GMP[19], and *myc* (Fig. 1h), both of which are expressed in *Platynereis* PSCs[18]. These data suggest that cluster 0 is a source of ectodermal PSCs.

As outlined above, *hox3* is preferentially expressed in ectoderm-derived PSCs. However, additional populations of stem cells contributing to *Platynereis* growth and regeneration have previously been hypothesized, including mesoderm-derived PSCs[21,42]. We therefore systematically queried our single-cell atlas with a combined signature of stem cells (*piwi, vasa, nanos*), proliferation (*proliferating cell nuclear antigen/pcna*) and chromatin remodeling (*dnmt1, chd1*) These genes are expressed in cells of post-injury timepoints within several clusters, hinting at additional sources of PSCs (Fig. 1g, h, Supplementary Fig. 3b–f).

To identify the most stem-like cells in each cluster in an unbiased, systematic way, we used CytoTRACE, a computational method which assigns cells a score representative of their "developmental potential", a proxy for stemness[43]. Cells were ranked by their CytoTRACE score (within each cluster), and genes correlated with this score were calculated. This analysis provides an unbiased, systematic overview of transcriptional changes within each cell population as cells acquire a

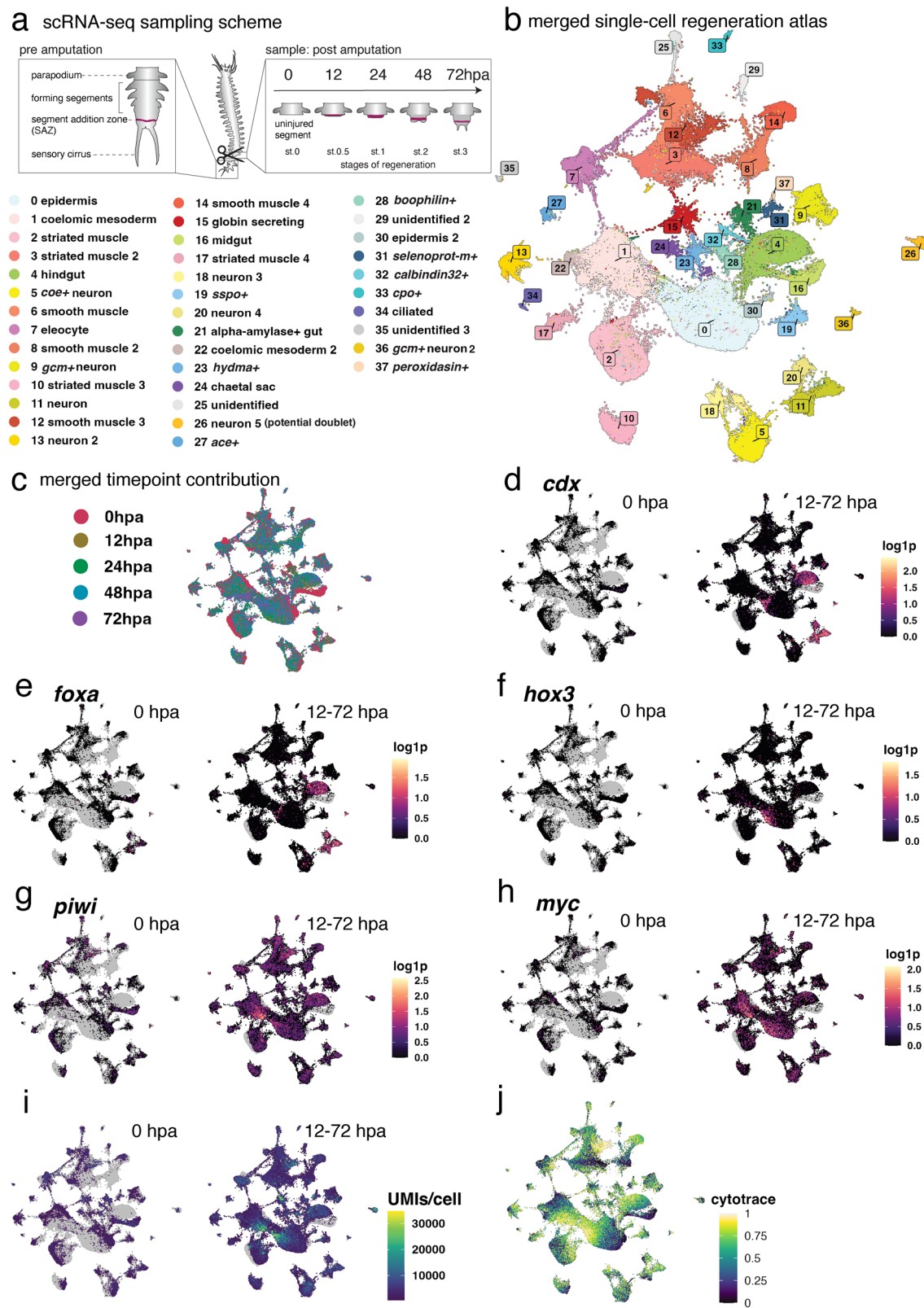

**Fig. 1 | A temporally resolved single cell atlas reveals the dynamic transcriptional landscape of cell populations during posterior regeneration. a** Sampling scheme illustrating posterior amputation and sampling timepoints, ranging from 0 hours post amputation (0 hpa, equivalent to a regular trunk segment) to 72 hpa, matching morphologically defined stages (st. 0 to 3). **b** UMAP visualization of cells, annotated by tentative cell type / population identity. **c** UMAP visualization showing the regenerative timepoint at which cells were sampled. **d**, **e** UMAP visualizations showing the expression of posterior identity markers (*cdx, foxa*) on the merged dataset, contrasting the freshly amputated sample (left) with the post-amputation time points (right) (**f**–**i**) similar UMAP visualizations, highlighting the changes in expression of stem-cell related genes (*hox3, piwi, myc*) (**f**–**h**) as well as the emergence of hypertranscriptomic cells (UMIs/cell) (**i**). **j** UMAP visualization of CytoTRACE values (calculated per cluster; high level indicates high differentiation potential); data for analysis provided as a Source Data file.

higher degree of developmental potential (Source Data). We further determined gene ontology (GO) terms associated with the transcriptional changes within each cluster, providing a more comprehensive resource for the involved biological processes (Fig. 1i; Supplementary Data 4 and below).

As an additional approach to identify potential stem cells, we took advantage of the observation that *Platynereis* PSCs exhibit larger nuclei and nucleoli[18,23], a feature usually associated with increased transcriptional (and translational) activity[44]. Increased, broad transcription, referred to as "hypertranscription", is frequently observed in active stem cells and progenitors, closely associated with proliferation, and plays a role in stem cell activation and function during growth and regeneration. Recently, absolute scaling of single cell transcriptomes using Unique Molecular Identifier (UMI)-based sequencing data has been shown to identify hypertranscriptomic stem cells and progenitors[45]. We therefore investigated the dynamic changes in transcriptional activity upon injury in our dataset and found evidence for hypertranscription (increased numbers of total UMIs detected per cell) (Fig. 1j). Our analysis shows that there is a progressive increase in high-UMI cells during regeneration (Supplementary Fig. 3g). Hypertranscriptomic cells are located within hotspots of GMP-related gene expression and high CytoTRACE values on the UMAP (Fig. 1i) and can be found in several clusters. Our analysis also revealed a sub-population of smooth muscle cells showing high CytoTRACE values and piwi expression even before injury (cluster 12, Fig. 1b, g,ii), which could imply the existence of a dedicated progenitor state within this specific tissue.

Taken together, we found several features associated with stem cells and the SAZ in injury-adjacent cells. These features are predominantly detected after injury, increasing as posterior regeneration proceeds. We found cells that strongly display these features distributed among multiple, but not all clusters in this dataset, indicating multiple different sources of regenerating PSCs.

To understand these putative sub-populations of the regenerated SAZ, and their cellular origins, we focused our analysis on two major cell populations (clusters 0 and 1), which show a strong activation of stem-cell-related features as described above.

To characterize these populations, we identified strongly expressed marker genes and performed in situ Hybridisation Chain Reactions (HCR, see Methods and Supplementary Data 5) to detect their expression in the tissue.[16,46]. Co-labeling of both genes in posterior parts of uninjured, posteriorly growing animals revealed that they demarcate two spatially distinct tissues, corresponding to the ectodermal epidermis (cluster 0) and a sub-epidermal mesodermal cell type (cluster 1) (Fig. 2a–e). The latter population covers the sub-epidermal region, but does not include muscle. We therefore refer to this population as coelomic mesoderm.

**Distinct signatures for PSCs of ecto- and mesodermal origin**
Having found evidence for distinct sources of regenerated PSCs, we next aimed to molecularly characterize them and profile them in situ. If indeed multiple populations of wound-adjacent cells acquire stem cell properties and repopulate the regenerating SAZ, our in silico data allows us to make certain testable predictions:

First, we examined whether cells of somatic origin change towards a teloblast-like morphology. As described above, *Platynereis* PSCs display a unique morphology with notably increased nuclear and nucleolar sizes. To test whether cells of this morphology emerge in wound-adjacent tissue, we stained tissue of posteriorly amputated *Platynereis* worms for the expression of *collagen alpha 6(VI) chain* (*col6a6*). Based on our CytoTRACE calculation, *col6a6* is strongly expressed in epidermal cells (cluster 0) and progressively lost as they acquire PSC-like properties (Supplementary Fig. 4a, Source Data). Quantifying the surface area of nuclei and nucleoli in this population during regeneration showed a strong increase in both metrics after

injury (Fig. 2f, g; Supplementary Fig. 4a–c, Source Data), along with a gradual reduction of *col6a6* levels. These data are consistent with the gradual acquisition of a teloblast-like morphology.

Next, we reasoned that if these PSC-like sub-populations are distinct from each other, we should be able to find genes specifically enriched in either of them and should find their expression in distinct groups of cells in situ. As mentioned above, *hox3* has previously been described as a marker predominantly expressed in ectoderm-derived PSCs, and accordingly is mostly restricted to the PSCs we identified among epidermal cells (cluster 0). Based on this observation, we sub-clustered cells of both the epidermal (cluster 0) and the coelomic mesodermal (cluster 1) populations to define their respective PSC-like sub-populations. We used CytoTRACE-scores, the total number of UMIs and the expression of GMP, SAZ, proliferation and epigenetic remodeling-related genes to identify the respective subclusters (Supplementary Fig. 4) and discovered novel molecular markers unique to these cells (Supplementary data 3; Supplementary Figs. 4d–f, 5a–s). These new markers include genes encoding putative receptors, as well as proteins with DNA binding motifs such as transcription factors, thus establishing a set of molecules with possible regulatory functions (Supplementary Data 3).

For ectoderm-derived PSCs, our analysis not only identifies the previously described genes *hox3* and *evenskipped (evx)* but adds markers such as a *gene encoding a fibronectin leucine-rich transmembrane protein of unclear orthology (flrtl,* Supplementary Fig. 6f–j) and a gene (*sp/btd*) encoding the *Platynereis* homolog of the transcription factor Sp9[47]. This population of cells further expresses early neuronal progenitor genes and patterning factors, such as the transcription factor gene *soxb1* (Supplementary Data 6) and the gene *four-jointed* that is involved in planar cell polarity, and was previously demonstrated to be expressed in developing medial neuroectoderm[36]. These data are consistent with the concept that these cells are the source of new neurons in post-regenerative growth.

For the mesoderm-derived population of PSCs, our analysis also predicts distinct marker genes. These include the gene *chd3/4/5b* that encodes a chromodomain helicase DNA-binding protein, and has previously been detected in regenerating mesoderm[24], as well as a gene we identify as *Platynereis paired-related homeobox* gene (*prrx*) (Fig. 2j, k, Supplementary Data 6; Supplementary Fig. 6k–o). The putative purinoreceptor gene *p2x* (Fig. 2i) and the *Platynereis* orthologue of the mesoderm related homeobox factor *msx*[48,49] are also predicted to be expressed in mesoderm-derived PSCs, albeit less exclusively than *prrx* (Supplementary Fig. 6k–t; Supplementary Data 3).

If *prrx* and *flrtl* are novel markers of distinct populations of stem cells, they should be expressed in separate, injury-adjacent populations of cells and exhibit morphological and molecular properties of stem cells. To test this prediction, we designed specific in situ HCR probes (Supplementary Data 5) and used these to analyze the expression of both genes in posterior regenerates (Fig. 2j, k; Supplementary Fig. 4d, e; whole-mount in situ hybridisation in Supplementary Fig. 4e). In agreement with our digital data, *flrtl* transcripts were co-expressed with *hox3* in cells of the epidermal layer, both at stage 1 and 3 (Supplementary Fig. 4d). By contrast, the predicted mesodermal stem cell marker *prrx* labeled cells at a deeper layer (Fig. 2j; Supplementary Fig. 4d). Consistent with the time-resolved atlas (Supplementary Fig. 6k–o), *prrx* was not yet detectable at stage 1 (Fig. 2l), but from stage 2 on (Fig. 2p). In both cases, a subset of labeled cells shows enlarged nucleoli as described for PSCs (arrowheads in Fig. 2h–k).

These sub-populations, based on our in silico data and their putative identity as stem cells, are predicted to be proliferating and expressing GMP genes. We therefore co-stained markers for ectoderm (*hox3*) and mesoderm (*prrx*) derived putative stem cells with the proliferation marker EdU and the key GMP factor *piwi*. We found both

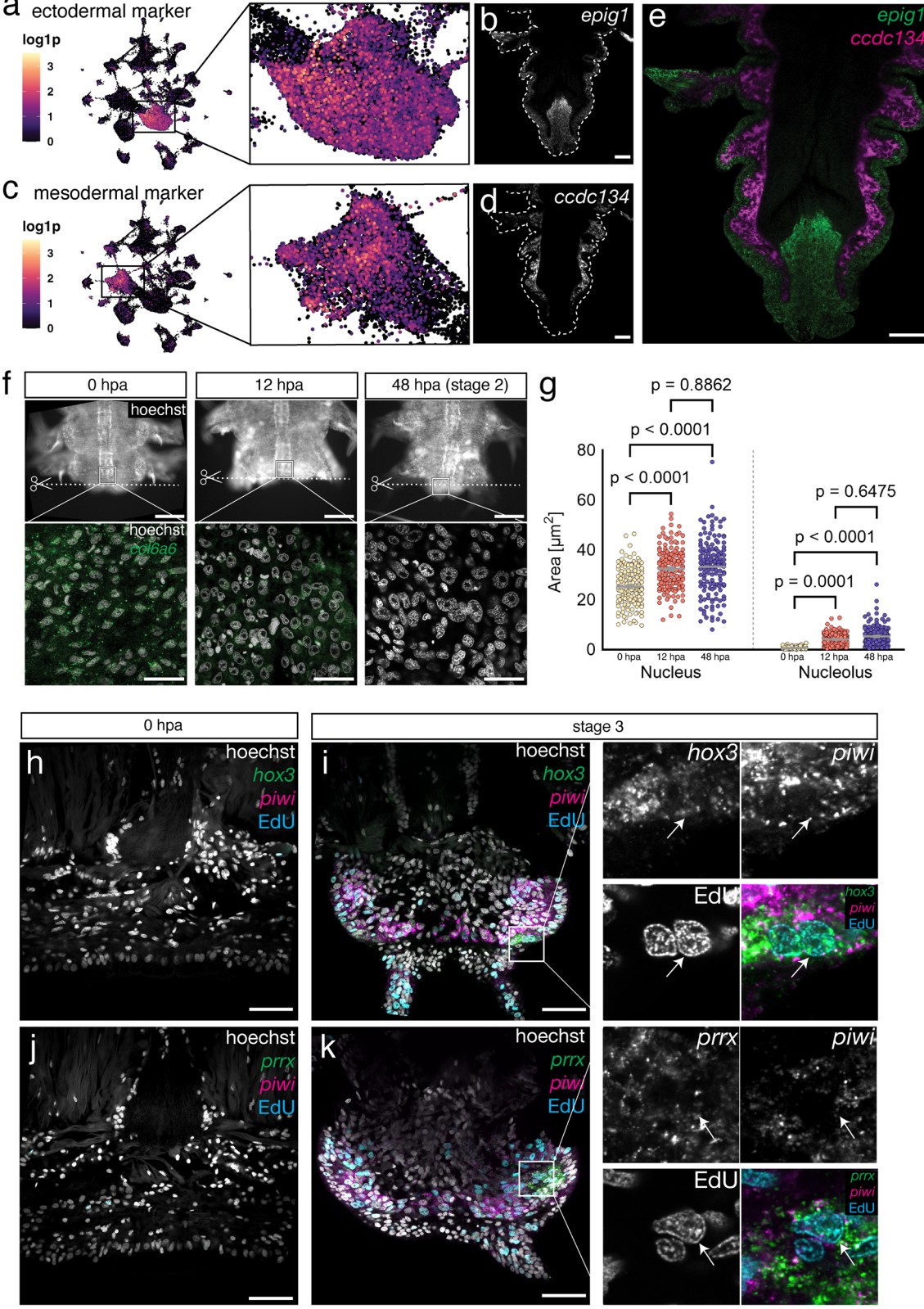

markers expressed in proliferating, *piwi* positive cells with teloblast-like morphology (Fig. 2h–j).

Taken together, these results are consistent with the notion that, during regeneration, distinct populations of PSCs of mesodermal and ectodermal origin derive from existing cells not displaying any stem cell related properties prior to injury. Our single-cell atlas allows the identification of novel markers of these cells.

## Germ-layer-based lineage restriction of growth and regeneration

Whereas our data argued for PSCs of distinct ecto- and mesodermal origin in the blastema, it still remained unclear if these cells had identical potency, contributing to derivatives in all of the regenerate, or if they were more restricted in their developmental potential. We therefore turned towards a transgenic strategy that

**Fig. 2 | Distinct wound-adjacent cell populations acquire stem cell properties upon amputation. a, c** UMAP visualization of markers specific to cluster 0 (*epig1*) and 1 (*ccdc134*). **b, d, e** in situ HCR (uninjured animal, posterior end), showing mutually exclusive expression of *epig1* and *ccdc134* in the epidermis and coelomic mesoderm, respectively. Scale bar = 50 μm. (*n* = 3–5). **f** Nuclear staining and in situ HCR of *col6a6* expression in wound-adjacent epidermal tissue at 3 timepoints after posterior amputation. Scale bar upper panels = 250 μm, lower panels = 25 μm. **g** Quantification of nuclear and nucleolar size change in regenerating tissue at 0, 12 and 48 hpa; each timepoint represents 3 individuals with 50 nuclei or corresponding nucleoli per individual. Statistical significance was calculated using a one-way ANOVA test with multiple comparisons. *P*-values for nucleus quantifications:

0pa vs 12hpa < 0.0001, 0hpa vs 48hpa < 0.0001, 12hpa vs 48hpa = 0.8862. *P*-values for nucleolus quantifications: 0pa vs 12hpa < 0.0001, 0hpa vs 48hpa < 0.0001, 12hpa vs 48hpa = 0.6475. Data for analysis provided as a Source Data file. **h–i** In situ HCR of regenerating tissue at 0 and 72 hpa (stage 3), showing the emergence of the expression of ectodermal stem cell marker *hox3*, combined with a ubiquitous stem cell marker (*piwi*) and a proliferation label (EdU, 30 min pulse before fixation); Scalebar = 50 μm. Zoom-in on *hox3* positive population. **j, k** In situ HCR of regenerating tissue at 0 and 72 hpa (stage 3), showing the emergence of the expression of mesodermal stem cell marker *prrx*, combined with a ubiquitous stem cell marker (*piwi*) and a proliferation label (EdU, 30 min pulse before fixation); Scalebar = 50 μm. Zoom-ins on *hox3-* or *prrx*-positive population.

would allow us to address this question at least on the broad level of germ layers.

Several other lines of evidence from previous studies suggest the existence of lineage-restricted ectodermal and mesodermal stem cells during larval and juvenile posterior growth in *Platynereis*[18,50–52]. The early embryogenesis of *Platynereis* follows a stereotypical programme known as spiral cleavage[53,54]. Highly asymmetric cell divisions (unequal cleavage) in the early embryo produce blastomeres of characteristic sizes and positions, whose fate are strictly determined[50]. Microinjection of a fluorescent lineage-tracing dye in individual blastomeres at the earliest stages of the spiral cleavage process, shows that ectodermal, mesodermal, and endodermal trunk tissues of the 4-day, three segmented larva are produced, respectively, by micromere 2d, micromere 4d, and the macromeres 4A-4D of the early embryo[50]. In another study, individual cells were tracked via live imaging from early embryogenesis into early larval stages to identify the fates of the mesodermal 4d blastomere[52]. This work revealed that the mesodermal bands and the primordial germ cells are produced by asymmetric divisions of the 4d lineage. In addition, the final divisions of the lineage during embryogenesis forms a group of undifferentiated cells at the posterior end of the hatched larvae, which will possibly become the mesodermal PSCs in later stages. Due to the transient nature of the signal (mRNA or dye injections), tracking the fate of putative PSCs into later juvenile stages was not feasible. However, molecular profiling[18] suggests but does not demonstrate the existence of at least two pools of PSCs with specific signatures, ectodermal and mesodermal, organized as two concentric rings anterior to the pygidium, the terminal piece of the *Platynereis* trunk (Supplementary Data 7, part A). So far, no transgenic lineage tracing technique has been used to clarify the origin of tissues in the posteriorly growing or regenerating juvenile.

To address this gap, we devised a mosaic transgenesis strategy using previously-established Tol2 transgenesis methods[15]. We constructed a Tol2 transgenesis construct with a nuclear mCherry and a membrane EGFP[52], under the control of the ribosomal protein rps9 promoter for ubiquitous expression[15] (Fig. 3a). We injected several batches of zygotes at the one-cell stage with the donor plasmid containing rps9::H2A:mCherry:F2A:mGFP transgene and transposase mRNA. These G0 worms typically show mosaic integration of the transgene. We raised the G0 batches that showed high numbers of surviving juveniles (Supplementary Data 7) (Fig. 3b–j). To screen these individuals for fluorescence patterns and identify which clonal lineages had the transgene integration, we amputated juvenile worms when they reached 6 weeks. These original tails (pygidium + a few growing segments) were imaged via confocal microscopy from both the dorsal and ventral sides. The amputated worms were further raised in individual containers and allowed to regenerate their posterior parts for three weeks. They were then amputated again one segment anterior to the regenerated part to collect the regenerated posterior parts for imaging. The whole cycle was repeated once. For each transgenic individual, we thus collected pictures of the primary clones derived from transgenic blastomeres as a result of normal development, as well as pictures of two reiterative, independent regeneration events

from the same primary clones originating from the non-regenerated trunk (Supplementary Data 7, part B).

Overall, we found that most individuals showed complex patterns of fluorescent primary clones. Although we cannot exclude that some of the patterns observed may be due to enhancer trapping, we see no indication that this phenomenon occurs significantly in our complete set of 62 transgenic individuals, presumably due to the relative strength of the ubiquitous promoter we have used (*rps9*). All six individual primary clonal patterns we deduce from observations are obtained multiple times (from 4 to 53 times, in 62 individuals, Supplementary Data 7, part C), practically excluding that they may be due to neighboring endogenous enhancers. The complexity of patterns likely results from a combination of reasons: Firstly, multiple blastomeres were transformed (Supplementary Data 7, part B, N09 and N35 for examples) resulting in combinations of tissues labeled. Secondly, only a part of a germ layer-derived tissue may be labeled. This is most evident in cases where only a bilateral half of the tissues is fluorescent because transgenesis happened in only one of the bilateral descendants of the germ layer founding blastomere (e.g. 4d divides bilaterally to give the precursors of the right and left mesoderm, Supplementary Data 7 part B, M24 and N23). Thirdly, some tissues were labeled in a stochastic, salt-and-pepper manner. This phenomenon is known as variegation[55] and presumably happens when a transgene is inserted near or within a heterochromatic region that imposes unstable transcriptional repression on it. This was particularly recurrent at the level of ectodermal tissues (Supplementary Data 7, part B, N25 and N33 for examples).

Despite this complexity, simpler patterns were also recovered in several individuals corresponding to the labeling of the whole trunk ectodermal tissues (Fig. 3b), the whole trunk mesodermal tissues (Fig. 3c) and the entire gut endoderm (Fig. 3e). The clonal nature of the ectodermal patterns is indicated by the continuity of expression of the transgene in PSCs, segmental precursors and differentiated segmental cells (Fig. 3f–i). Ectodermal PSCs, corresponding in location and cytological characteristics to the ring of cells identified by molecular signature before (Gazave et al.[18]), are easily identifiable (Fig. 3h). Potential mesodermal PSCs are also tentatively imaged in locations already identified molecularly (Supplementary data 7, part B, M03). These primary clones support the aforementioned concept that separate pools of precursor cells generate these sets of tissues during the life-long process of posterior addition of segments. As for the endoderm, so far, no endodermal PSCs have been identified by molecular signature, and it is possible that endodermal precursors or stem cells are spread in a diffuse way along the length of the trunk[56].

In addition to the trunk germ layer-derived tissues, several primary clonal patterns were obtained repeatedly either alone or in combination with others (Supplementary Data 7). The pygidial ectoderm was often labeled independently of the trunk ectoderm (Fig. 3d). This demonstrates that the pygidial ectoderm is derived from blastomeres different from the trunk ectoderm and that the anterior border of the pygidial ectoderm with the trunk ectoderm is a compartment border with no contribution of the pygidial cells to the

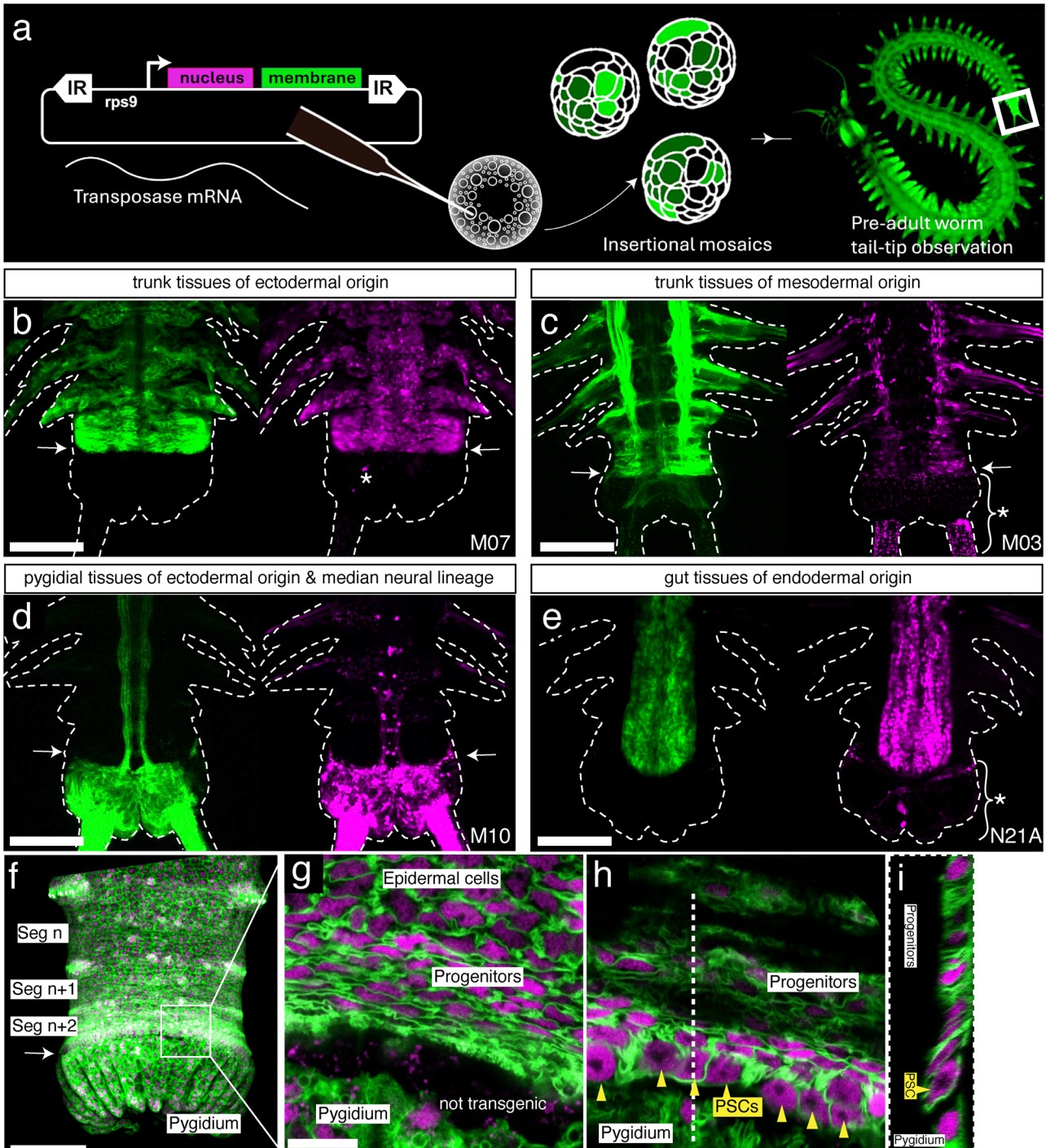

**Fig. 3 | Mosaic transgenesis reveals developmental compartment restrictions of PSCs in *Platynereis* posterior elongation. a** Summary sketch of the protocol for creating large embryonic clones with transposase. **b–g** Examples of simple germ layer– or tissue-specific primary clones observed on posteriorly growing worms; 6 weeks, ventral views. Observed frequencies of integration patterns for ectoderm = 29/61, mesoderm = 6/61, pygidial ectoderm = 53/61, median neural = 13/61, endoderm = 16/61 (Supplementary Data 7). **b** Ectodermal clone, no pygidial or internal cell labeled. **c** Mesodermal clone, only the muscles are clearly visible. **d** Pygidial and median neural lineage clones. Most median neurites are emanating from pygidial sensory neurons. **e** Endodermal clone. **f–i** Dorsal views of a primary clone in ectoderm-derived PSCs. **f** General confocal stack projection, showing the position of the ectoderm-derived PSCs and uniformly labeled nascent segments; the pygidium is labeled with independent clones. Magnified confocal section views of the SAZ region, at 2 μm z-depth (**g**), 6.5 μm z-depth (**h**) and and y-z section (**i**). **g–i** show the continuity of the clonal expression of the transgene in bottleneck-shaped ectoderm-derived PSCs with large nuclei-nucleoli (yellow arrowheads), transversely elongated columnar progenitor cells and squamous epidermal differentiated cells. For all panels, green labelings are cell membranes, magenta labelings are cell nuclei. White arrows: position of the PSCs. White asterisks: background staining. Scale bars **b–f** = 100 μm; **g–i** = 10 μm.

growth of the trunk ectoderm (Fig. 3b, d). The one exception to the pygidium/trunk compartmentalization is the presence of a median neural lineage (Fig. 3d), composed of two pairs of cells per new segment, that is also identified alone, sometimes unilaterally (Supplementary Data 7, part B, M09, M10, N04, N11, N14, N26, N40 and O15). These cells are probably produced by independent median specialized posterior stem cells that segregate from the 2d lineage in the early embryo. Lastly, a lineage of amoeboid, presumably phagocytic cells, possibly derived from anterior embryonic mesoderm, was observed several times (Supplementary Data 7, part B, M09, N03, N05, N22).

Most importantly, germ-layer compartmentalization is fully conserved during regenerative events (Fig. 4), with each germ layer of the regenerate originating exclusively from cells of its kind in the neighboring non-regenerated trunk. The clonal nature of the transgene expression is again illustrated by the continuous transgene expression in the differentiated epidermal cells, blastemal cells, and the regenerated PSCs (Fig. 4a–f). Ectodermal regenerated PSCs are clearly identifiable as soon as 4 days post amputation (Fig. 4c, d). Lineage restrictions in the regeneration blastema (Fig. 4g–l) are in agreement with the distinct source populations of stem cells suggested by our scRNAseq analyses. A diagram of the whole set of primary clones obtained fully supports this interpretation of compartmentalization (Supplementary Data 7, part C).

While these transgenic clones do not demonstrate the embryonic germ layer origins, they show that tissues remain strictly compartmentalized during posterior segment addition, similar to embryonic/larval development. Taken together, all these results are compatible with the presence of the two rings of ectodermal and mesodermal PSCs immediately anterior to the pygidium/trunk border, while the unsegmented endoderm may grow diffusely or through the activity of specific endodermal PSCs yet to be identified. After amputation (which removes all PSCs), ectodermal and mesodermal PSCs, as well as endoderm, are regenerated exclusively from precursors of their kind in the uncut segments, either from dedifferentiating cells, or from unknown resident lineage restricted precursor cells, in complete agreement with the single-cell transcriptomics clustering.

### Activation of PSCs and regeneration requires TOR signaling

The protein kinase Target Of Rapamycin (TOR) has been implied in wound response and blastemal signaling in planarians, zebrafish and axolotl[57–61], reviewed in ref. [62]. Our in silico GO-analysis revealed increased expression of TOR- related transcripts in cells as they acquire PSC identity in response to injury in *Platynereis* (Supplementary Data 4). These include the *Platynereis* orthologs of genes encoding TOR (Supplementary Data 6) as well as components of the ragulator complex (lamtor 1, 2, 3, 4 and 5), which is involved in TOR complex activation and localisation, and therefore might influence cell metabolism and proliferation[63]. Additionally, many biological processes known to be controlled by TOR activity were found enriched in our GO-term analysis of genes associated with high CytoTRACE scores (e.g. translation, rRNA processing, ribosome biogenesis, see Supplementary Data 4).

Moreover, increased TOR signaling has also been observed in hypertranscriptomic cells[45], as we observe them in the *Platynereis* regeneration process, and TOR complex activity has been shown to be a key requirement for maintaining a hypertranscriptomic state in embryonic stem cells[64]. We therefore investigated whether posterior regeneration in *Platynereis dumerilii* also required a functional TOR signaling system.

In our single-cell atlas, *Platynereis tor* was broadly expressed, including in the tentative PSC subpopulations (Fig. 5a–c). To assess whether or not TOR signaling was required for regenerating PSCs after amputation, we treated amputated animals with the ATP competitive TOR inhibitor AZD8055[65] and compared their regenerative success to DMSO-treated controls (Fig. 5d–l). Already at 24 hpa, so before the

formation of a significant blastema or a strong increase in cell proliferation[21], in situ HCR revealed that *myc* expression was strongly reduced and *hox3* expression was completely undetectable in treated animals (Fig. 5e, f), whereas control animals successfully established a zone of *myc*+ and *hox3*+ cells (Fig. 5i, j). Additionally, treated animals did not reach a stage 3 regenerate at 72 hpa (Fig. 5g, h). DMSO-treated control animals progressed normally and regenerated a blastema and early developing anal cirri within the same timespan (Fig. 5k, l). We therefore conclude that, upon TOR inhibition, *Platynereis* fails to regenerate PSCs, and subsequently does not develop a blastema or differentiated posterior tissues.

## Discussion

As outlined above, our work is consistent with the classical proposal that formation of the regenerative blastema of *Platynereis dumerilii* involves a process of injury-induced "re-embryonalisation" of wound-adjacent cells[23]. We advance this model to cell-type resolution, find molecular similarities to vertebrate blastema formation, and present a clonal analysis in 62 individuals, furthering our understanding of lineage restriction during this process. Our data are consistent with both morphallactic repatterning processes and multiple parallel dedifferentiation events, and provide the molecular fingerprint of distinct groups of stem cells in posterior regeneration. These findings complement observations based on candidate genes and transcriptomic analyses in *Platynereis* and other annelid systems[21,66], and offer a fresh perspective on fundamental regenerative processes.

Both our transcriptomic and clonal analyses argue that, unlike planarians that exhibit pluripotent stem cells capable of regenerating all cell types of the adult[67], *Platynereis* regeneration relies on cells with limited potency that respect the distinction between cells arising from different germ layers in development. In both of these spiralian species, however, there appears to be a continuity of potency mechanisms between normal growth and regeneration processes: In planarians, pluripotent neoblasts are not only relevant for regeneration, but also homeostasis and growth[68]. Likewise, in *Platynereis*, lineage restriction applies to transverse growth, posterior growth, and regeneration. The dissimilarity of stem cell potencies within the clade of spiralians is reminiscent of the diversity of growth and regeneration mechanisms also found in cnidarians, where pluripotent stem cells are found in *Hydractinia*[69], whereas *Hydra* employs lineage-restricted progenitors[70]. As single-cell analyses in other annelid species are becoming available[66,71], we expect that the availability of time-resolved cellular and molecular data in the *Platynereis* model will help to more easily delineate differences and commonalities for regeneration-relevant stem cell mechanisms also in other spiralian species.

A recent single-cell study on the stem cell system in *Pristina leidyi*[71] found evidence for *piwi*-positive stem cells spread throughout the adult body of this annelid. The analysis identified a single, potentially pluripotent pool of stem cells at the root of all adult tissues. However, transcriptional heterogeneity was detected in this *piwi* positive population, and lineage-restriction could not be ruled out. Unlike *Platynereis*, *Pristina* reproduces by fission, and a large population of *piwi* positive cells was detected in a developing fission zone. As the experiments in *Pristina* were not conducted in a regeneration context, further experiments will be needed to allow direct comparisons of stem cell systems between these two more closely related species. Such comparisons could reveal important insights into the evolution of asexual reproduction and how it affects stem cell potency and availability.

As the labeled clones we obtain by zygotic injections are primarily large, they are well suited to provide a clear view of germ-layer-restricted lineages, but cannot yield experimental access to smaller lineages in both posterior growth and regeneration. Our single-cell transcriptomic atlas corroborates multiple distinct clusters of

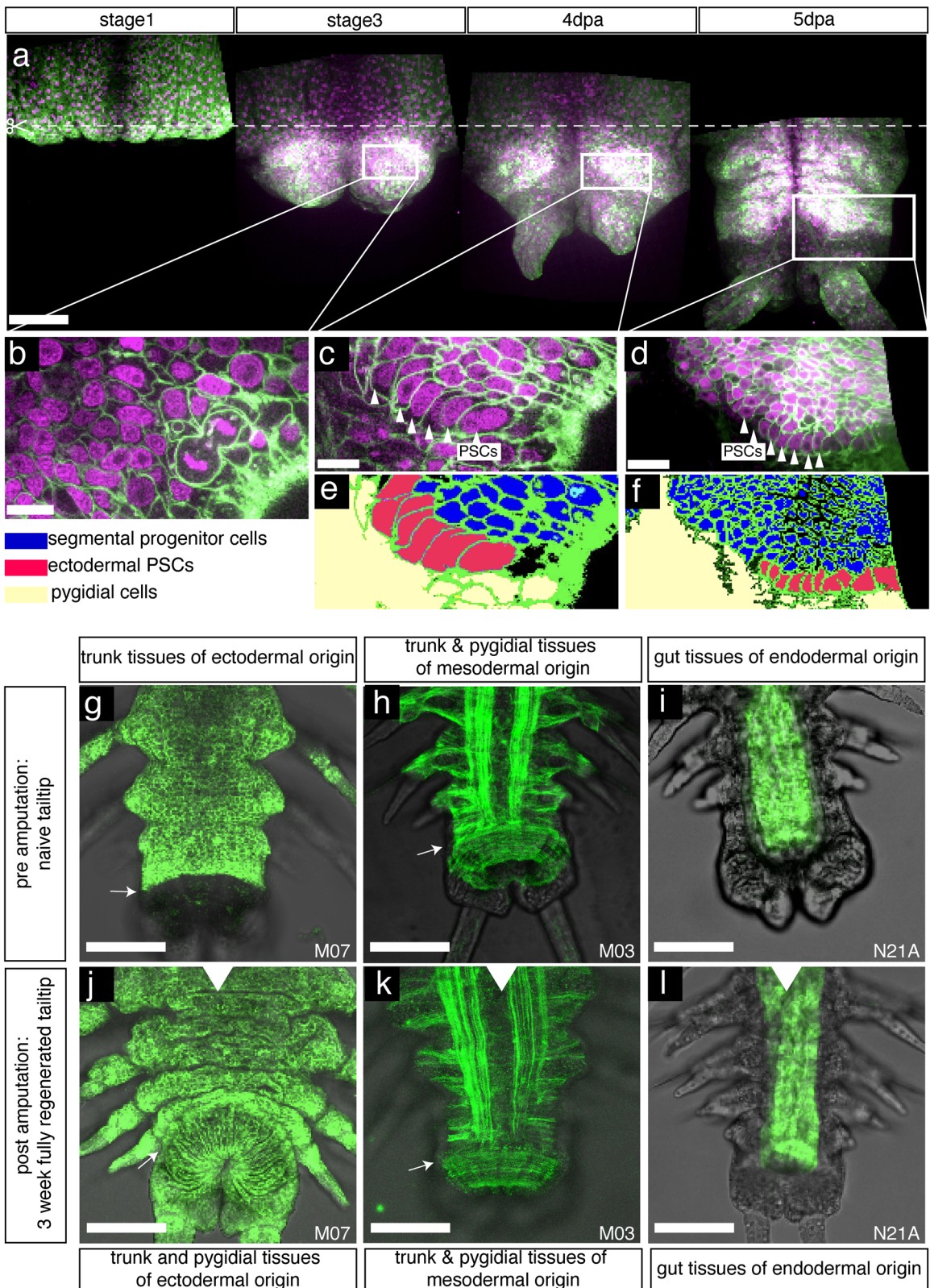

segmental progenitor cells
ectodermal PSCs
pygidial cells

progenitors that are confined to individual germ layers, but also exhibit further subdivisions. It is thus possible that there are even more restricted lineage compartments currently not accessible to experimental validation. In turn, not every transgenically labeled compartment contributes to its regenerated counterpart. One example is the bilateral neural lineage that we identify, which is continuous with the pygidium before amputation, but does not visibly contribute to the

regenerate. The existence of such a developmental compartment is consistent with the idea that there might be distinct subsets of *hox3*-positive PSCs in the ring-like segment addition zone, possibly reflecting the existence of distinct neurogenic columns in posterior growth[72]. More refined mapping techniques will be required to assess if such subsets exist in regular development, and if they are reconstituted in the process of regenerative growth.

**Fig. 4 | Persistent labeling of cells within developmental compartments during regeneration. a** Time lapse ventral confocal stack projections of the regenerating tail tip of a worm displaying clonal transgene expression in the ectodermal lineage. **b–d** magnified confocal sections of the same individual. **e, f** interpretative schemes of **c** and **d**. The time-lapsed views illustrate the continuity of clonal expression of the transgene in epidermal cells (**a**, stage 1), undifferentiated blastema cells (**b**), regenerated PSCs (**c, d**) and progenitor cells (**c, d**). **g–l** Regeneration experiments on animals bearing simple clones. Dorsal views of confocal stack projections, with pre-amputation views on top and the matching full regenerates (3 weeks post amputation) on the bottom. This series illustrates the strict compartment restriction in the regeneration of ectoderm-derived and mesoderm-derived PSCs, as well as gut endodermal lineage. Pygidial ectoderm, entirely removed upon amputation, is regenerated exclusively from trunk ectoderm precursors (**g, j**). For all panels, green labelings are cell membranes, magenta labelings are cell nuclei. White arrows: position of rings of PSCs in the respective focal plane. Scale bars **a** = 100 µm; **b, c** = 10 µm; **d** = 20 µm; **g–l** = 100 µm;.

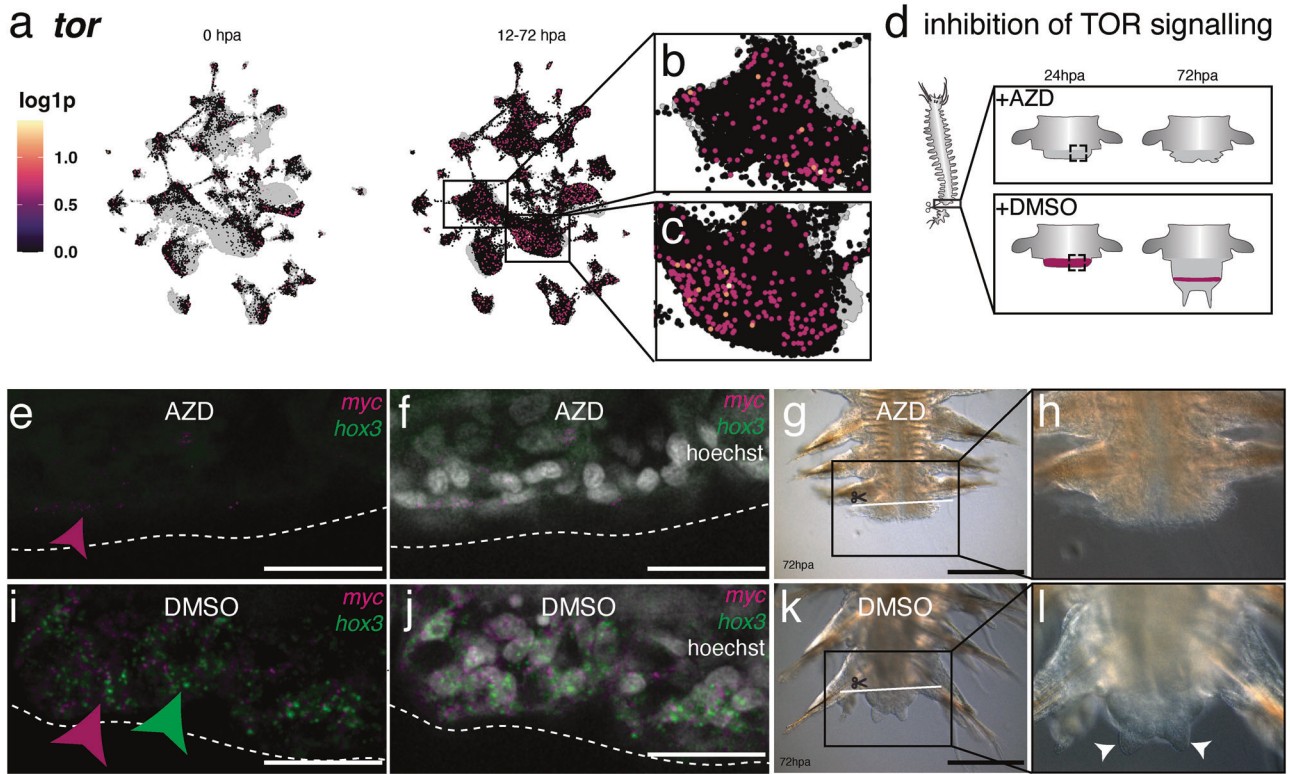

**Fig. 5 | TOR signaling is necessary for re-establishing stem cell gene expression profiles and morphological regeneration upon injury. a–c** UMAP visualization of *tor* expression; **a** comparison between 0 hpa and 12-72 hpa; **b, c** enlarged views of cluster 0 and cluster 1 in 12–72 hpa samples. **d** Scheme of posterior amputation, TOR inhibition and posterior regeneration after 24 and 72 h, highlighting region used for assessing stem cell gene expression 24 hpa. **e–h** Analysis of amputated animals (n = 6) treated with AZD8055 TOR inhibitor at 24hpa and 72hpa. **e, f** Confocal images of in situ HCR stainings detecting expression of *hox3* and *myc* at 24 hpa; **g, h** brightfield images at 72 hpa; **i–l**. Equivalent analyses in DMSO-treated controls. (n = 6) (**i, j**) Confocal images of in situ HCR stainings for *hox3* and *myc* at 24 hpa; **k, l** brightfield images of posterior regenerates at 72 hpa. Scale bars = 25 µm (**e, f, i, j**); = 250 µm (**g, h, k, l**).

Our data not only provide insight into the likely source and restriction of PSCs in *Platynereis* regeneration, but also into molecular factors involved in the emergence of blastemal cells and their possible conservation. Based on our findings and existing literature, we propose a working model of regeneration in *Platynereis* consisting of the activation and dedifferentiation of wound-adjacent cells, their acquisition of distinct, lineage restricted stem cell properties and ultimately their proliferation and contribution to regenerating tissue. While dedifferentiation as a source of blastemal stem cells in annelids has previously been proposed[8,10,20,21,23,25], we report cell-based molecular data consistent with this concept. We observed multiple distinct populations of cells responding in similar ways: they start expressing *myc* which, besides its general role in stem cell activity, has been shown to be involved in mammalian dedifferentiation and pluripotency[73,74], as well as several genes of the germline multipotency programme (e.g. *piwi, vasa, nanos*), which are also expressed in *Platynereis* PSCs during regular growth. These cells show hypertranscription, a known feature of active adult stem cells[44,45], strong expression of genes involved in epigenetic reprogramming (e.g. *chd3/4/5b, dnmt1, chd1*), and start to proliferate. Both

*myc* and *chd1* have previously been shown to play a central role in stem cell activation and hypertranscription in many species, such as the endothelial to hematopoietic stem and progenitor transition in mouse development[44,75].

Together, these findings strengthen the argument for multiple, parallel dedifferentiation events underlying regeneration in *Platynereis*. Our data do not exclude the possibility that resident stem cells or dedicated progenitors among heterogeneous pools of differentiated cells could also contribute to the regenerate, as has been proposed or observed in multiple regeneration models including annelids[5,8,10]. Our findings of *piwi* expression and high CytoTRACE value in a subset of smooth muscle cells before injury might indicate the existence of a progenitor-like cell state in this particular tissue. However, the fast de novo establishment of the molecular stem cell signature within 12–24 hpa, before a detectable increase in proliferation commences, argues that such a contribution may only account for a limited set of blastemal cells, or specific tissues, as suggested for gut cells during regeneration in a recent publication[76].

Apart from those more general processes involved in the regeneration programme, the notion that PSCs resulting from epidermal

and coelomic mesodermal cells are molecularly distinct will also be helpful in more clearly delineating possible parallels to established, dedifferentiation-based regeneration mechanisms in other systems. Indeed, the expression of the *sp9* homolog *sp/btd* in epidermis-derived PSCs of *Platynereis* is reminiscent of the expression of *sp9* in the dedifferentiating epidermis of axolotl blastemas[77]. In turn, similar to the axolotl *msx-2* gene that becomes refined to the mesenchymal part of the blastema[78], we identify its *Platynereis* ortholog *msx* to be present in the bona fide mesodermal PSC clusters. Likewise, our work revealed a previously uncharacterized *Platynereis prrx1* orthologue that is expressed in the mesodermal PSCs. Vertebrate *prrx1* genes are prominently expressed in mesenchymal cells during limb development of chicken[79], and mouse[80], and have also been characterized as part of the connective tissue progenitor cells in axolotl and frog blastemas[22,78,81]. It has previously been argued that evolutionary understanding of blastema-based regeneration will require a more detailed understanding of the underlying genetic circuitry[3]. Our work establishes *Platynereis* as a promising candidate for such analyses and provides a first comprehensive set of data towards this end.

The prevalent expression of ribosome- and cellular growth related genes in PSC-like cells during regeneration warranted an investigation of the role of TOR signaling in the activation or dedifferentiation or activity of stem cells in our model. As discussed above, TOR signaling has previously been implied in regeneration in many species, with a recent article demonstrating its crucial, regulatory role upstream of axolotl limb regeneration[57]. While the exact role this pathway plays in this process is not well understood, its activity seems to be generally required for regenerative stem cell proliferation. Interestingly, the formation of stem cells through dedifferentiation occurs during a period of lower TOR activity and high autophagy (which TOR usually inhibits)[62]. We found that impaired TOR signaling in *Platynereis* not only affects proliferation and the morphological formation of a blastema, but directly blocks the otherwise reliable establishment of PSCs as early as 24 hpa, suggesting an early, central role of TOR kinase activity in regulating the dedifferentiation of cells in response to injury.

Lastly, we expect our methodological advances presented in this manuscript to be of broader use in the establishment of additional resources for comparative regeneration biology. Using a widely applicable cell fixation method and parallel processing of all samples allowed us to generate a merged, temporally resolved dataset without having to rely on computational batch effect removal strategies. Similarly, whereas regular pre-processing would normalize cellular transcriptomes to comparable levels, we suggest that the hypertranscriptomic state of cells at later regenerative timepoints reflects a biological feature of PSCs. Finally, our clonal analysis approach offers a direct way to observe clonogenic lineages in development and regeneration as well as new insights into *Platynereis* lineage restriction, without requiring stable expression of transgenic constructs beyond G0. Taken together, our manuscript therefore provides a framework and methodological toolkit for future projects aimed at acquiring and analyzing the data required to compare regeneration across species and ultimately advance our understanding of blastema-based regeneration.

## Method
### Animal culture
*Platynereis dumerilii* were kept in laboratory culture at temperatures between 18° and 20 °C in a 16:8-h light-5 dark (LD) cycle, and maintained under previously published conditions[17,82], adhering to the applicable national legislation that does not require a separate ethical approval.

### Posterior amputation surgery
To perform posterior trunk amputation surgery, animals were anesthetized in 7.5% MgCl₂ diluted 1:1 with sea water. For bulk- and single cell transcriptomics and for all in situ HCR labelings shown, animals of 40–60 segments size and 3–6 months age showing no signs of sexual maturation and no prior injuries were sampled. Amputations were done by performing a transverse section posterior of segment 30, using surgical scalpels (Swann-Morton, Type 11). For regeneration time courses, animals were then rinsed in sea water and transferred to fresh culture boxes for recovery.

### Bulk RNA sequencing of regenerating tissue
Posterior surgeries were performed as described above. Tissue was then harvested at each sampling time point by anesthetizing animals again as described above, followed by a transverse cut anterior of the wound-adjacent segment, right at the segment boundary. Tissue pieces were transferred to sea water, pooling 8 pieces per replicate, on ice. After tissue pieces sank to the bottom of the reaction tube, the supernatant was removed and the samples were frozen in liquid nitrogen and stored at −80 °C.

RNA extractions were performed using a commercial kit (Directzol RNA MiniPrep, Zymo Research, USA), following the manufacturer's guidelines. RNA was eluted in 30 μl RNAse-free H₂O. Illumina sequencing libraries were prepared at the Vienna Biocenter Next Generation Sequencing Facility, using NebNext Dual adaptors with dual indexing. Libraries were then sequenced using the Illumina NovaSeq platform with S1 PE100 flowcells.

### Bulk RNA sequencing analysis
Sequences were trimmed using cutadapt (v. 1.12)[83] and analyzed for sequence quality using fastQC (v. 0.11.9)[84] and multiQC (v. 1.14)[85]. We used STAR aligner (v. 2.7.10b)[86] to generate a reference file from the *Platynereis* genome assembly draft (*genomeGenerate*) and align sequencing reads (*alignReads*), then extracted a counts matrix using featurecounts (v. 2.0.1)[87]. Data were then processed using DESeq2 (v. 1.36.0)[88] and mfuzz (v. 2.56.0)[89], estimating ideal clustering variable and cluster number following the software package guidelines. Cluster members (membership cutoff = 0.5) were then plotted using ggplot2 (v. 3.4.2)[90], with select genes of interest plotted in color (Supplementary Fig. 1).

### Single cell transcriptome sequencing
For single-cell RNA sequencing, we performed posterior amputations and regeneration time courses similar to those described above. However, after sampling the wound-adjacent segments, we transferred those to an acetic acid/ methanol (ACME) solution with 1% NAC for dissociation[27]. Cells were then dissociated over the course of 45 min, interspersed with rigorous pipetting every 15 min using a P1000 pipette pre-coated in 1% BSA in PBS to reduce cell loss due to stickiness. Dissociated cells were centrifuged (1000 g for 5 min at 4 °C), resuspended in storage buffer (1% BSA in PBS, with 10% DMSO and 1U/μl recombinant RNAse inhibitor, Takara Bio) and stored at −20 °C for further processing.

To remove debris and concentrate cells, we performed FACS on freshly thawed samples. We first labeled nuclei (Hoechst 33342 at 5 μg/ml, for 15 min at room temperature), then sorted 15,000 cells directly into 10X genomics chromium buffer, using FACS gates (FSC-A vs FSC-H; DAPI-A) to exclude debris and clumped cells (BD FACS Aria IIIu).

Single cell barcoding droplet (GEM) production and library preparation was performed at the Vienna BioCenter Next Generation Sequencing facility using the 10X genomics Chromium platform according to manufacturer's instructions (10×3' v3 chemistry, 10X Dual Index Kit). Two independent experiments were performed (replicates a and b), following the same strategy: all animals for one experiment were sampled from sibling- or closely related batches of animals. Amputations were performed in parallel, and by freezing ACME-dissociated cells, we were able to store sampled cells throughout the regeneration time course. All samples were then sorted, and barcoded

in parallel. Libraries were sequenced on the Illumina NovaSeq platform (S4 lanes, PE150).

## Single cell data processing

We used Cellranger software by 10X genomics (v. 7.0.1) to generate a custom genome reference (*mkref* function) from the *Platynereis* genome (Genbank ID: GCA_026936325.1), using gene annotations provided in ref. 91. We then assigned reads to cellular barcodes, aligned them to the custom reference and generated a read barcode matrix for each sample individually using the *cellranger count* function with *expect-cells* set to 10,000.

Our main processing pipeline was based on the "Seurat" package for R[92], with specific modifications for merging multiple datasets. In brief, we imported the barcode matrices (*Read10X* with min.cells = 3 and min.features = 200) and followed standard pre-processing steps. Outlier cells were removed based on manual inspection of scatterplots (counts vs features), removing between 30 and 108 cells per dataset. Quality metrics and cutoff values for all samples are available in (Supplementary Data 1).

We merged all datasets following an approach employed elsewhere[93]. We normalized the data (LogNormalize with scale-factor = 10,000) and identified the variable features for each dataset individually (using FindVariableFeatures). We then used the union of all variable features from all different timepoints to conserve features which might be variable only at a certain timepoint, and used this set of features for the merged dataset. Next we scaled the merged dataset and performed dimensionality reduction (first 50 dimensions in PCA space). We then calculated the UMAP embedding and performed cell-type clustering. After trying multiple clustering resolutions we settled on a resolution of 0.5 as it best reflected known biological cell types and subpopulations.

## Single cell cluster annotation

Marker genes were identified for each cluster using the Seurat *FindAllMarkers* function (only returning positive markers at a min.pct of 0.25 and a log-fold change threshold of 1). As the reference genome used in this manuscript lacks a gene name annotation, we annotated marker genes here and below using a table of best BLAST hits for each coding sequence identified on the genome (see "transcriptome annotation" below). The annotated table of markers for each cluster (Supplementary Data 3), combined with screening expression of an array of genes with known expression patterns in *Platynereis* (Supplementary Data 2) were then used to annotate the clusters.

## Bulk- and single cell sequencing data comparison

For quality control, we correlated single cell and bulk RNA sequencing data. Due to the differences in sequencing technologies, strong correlation is not necessarily expected. However, mismatched correlation between bulk- and single cell timepoints might indicate biological differences or technical issues. To this end, we extracted aggregate ("pseudobulk") counts from the single cell dataset (grouped by timepoint) using the Seurat AggregateExpression function. Bulk sequencing counts were extracted using the DESeq2 counts function. All counts were then subset to only contain features identified as variable in the single cell object, expressed in at least one replicate in both bulk and single cell data. We then scaled all remaining features by division by the sum of each feature's expression. Pearson correlation was calculated using the cor function (stats package v4.3.3) with default parameters and visualized using the pheatmap package (v1.0.12) function pheatmap, also at default parameters.

## Single cell doublet prediction

Potential cell doublets (2 or more cells assigned the same cellular barcode) were estimated using the R package DoubletFinder (v2.0.3 - note that the latest version, 2.0.4., introduces a code-breaking change)[29]. In brief, optimal parameters were determined and doublet scores were calculated for each dataset independently. Expected number doublets based on fluid dynamics in 10X Chromium devices was determined based on manufacturer instructions (https://kb.10xgenomics.com/hc/en-us/articles/360059124751-Why-is-the-multiplet-rate-different-for-the-Next-GEM-Single-Cell-3-LT-v3-1-assay-compared-to-other-single-cell-applications), rounding to the closest available number. For doublet prediction, 30 principal components and a pN value of 0.25 were used.

## Analysis of UMIs per cell

UMI values of cells were compared between timepoints post amputation using the Wilcoxon rank sum test with continuity correction (Supplementary Fig. 3). To ensure these results were not affected by outlier removal (see above), we performed the same test on cells without removing outliers, yielding comparable results (0 hpa vs 24 hpa: $p < 2.2e{-}16$; 24 hpa vs 72 hpa: $p < 1.227e{-}11$).

## CytoTRACE analysis

To rank cells by their developmental potential, we used CytoTRACE (v.0.0.3). CytoTRACE is a statistical method, which uses transcriptional diversity as a proxy for developmental potential and assigns a CytoTRACE-score to each cell[43]. CytoTRACE scores were calculated for each cluster independently, using default parameters, then transferred to a metadata slot of the Seurat object. Several biologically similar clusters were merged for this analysis (3, 6 and 12; 4 and 16; 8 and 14). While CytoTRACE has been shown to work on a broad range of datasets and organisms, we focussed our analysis on clusters in which known biological information (e.g. expression of established stem cell marker genes) could be used to assess its results. Data for all clusters is available (Supplementary Data 4), and validations (i.e. CytoTRACE score matching expression of stemness-related genes) were performed for clusters 0 and 1.

## Gene annotation

Top genes of every cluster were annotated using an automated pipeline. For this, all transcript sequences predicted from a given gene locus (XLOC ID) were used for sequence searches using BLASTX[94] against two protein databases: a version of the NCBI Uniprot/Swissprot repository (accessed on June 16, 2021) and a database combining entries of the more inclusive NCBI RefSeq repository (accessed on November 25, 2021). For the latter one, all protein sequences available for a set of representative landmark species and taxa were used: Annelida, the mollusks *Mytilus galloprovincialis, Crassostrea gigas, Mizuhopecten yessoensis, Octopus bimaculoides, Pomacea canaliculata, Lymnaea stagnalis, Biomphalaria glabrata, Aplysia californica*; the insects *Drosophila melanogaster, Apis mellifera, Clunio marinus, Nasonia vitripennis*; and the chordates *Branchiostoma, Mus musculus, Oryzias latipes, Danio rerio* and *Gallus gallus*. Best hits for each of these searches were tabulated. For gene loci with multiple transcripts, the results for the transcript that retrieved the highest score in the analyses was retained as reference, so that each gene locus retrieved one annotation.

For investigation of specific genes, we assembled bona fide full length sequences independently of the genome annotation, using available RNAseq data generated for the laboratory strains (PIN, VIO) maintained in the laboratory[9]. Sequences from individual libraries were assembled using the CLC Main Workbench Software package (version 23.0.2), and predicted protein sequences subjected to domain analysis using SMART[95]. Assembled gene sequences for *Platynereis prrx, flrtl, p2x, epig1, ccdc134, tor, lamtor1 to 5*, and *smg1* were submitted to the NCBI Genbank repository.

## Gene Ontology analysis

To assess the CytoTRACE-scores further and look for potentially informative associated gene expression patterns, we calculated

Pearson-correlation values for each gene with the CytoTRACE score of each cluster, using the CytoTRACE software with default parameters[43]. We then ordered genes by this correlation and performed GO term enrichment analysis for each cluster using CERNO[96] as implemented in the tmod R-package (version 0.50.13)[97].

Pearson correlation scores with CytoTRACE scores were calculated for the full gene-set. Because GO terms are not annotated for *Platynereis*, we identified the closest matching human gene symbol (see "gene annotation" above, limited to human gene hits) and translated these to entrez-ids for GO term analysis. Therefore, this analysis exclusively focuses on genes with a close human orthologue. We then used the three human GO term annotation sets as accessible through the org.Hs.egGO R-package (version 3.17.0) and performed the CERNO enrichment test.

### Phylogenetic analyses
For phylogenetic analyses of selected *Platynereis* proteins whose phylogeny was not yet previously reported, we used the following strategy: We identified the top hits in selected reference species representing key vertebrate phyla (mammals, birds, amphibians) as well as invertebrate phyla (mollusks, insects) by performing BLASTP searches against the NCBI clustered nr database. For identifying/completing matching sequences in the axolotl (*Ambystoma mexicanum*), we made use of the Axolotl-omics web resource (https://www.axolotl-omics.org/) that allowed access to the latest axolotl transcriptome assembly (AmexT_v47). Proteins were aligned using the CLC Main Workbench Software package (version 23.0.2) and cleaned of short sequences. Subsequently, alignments were exported and used for phylogenetic analyses using IQ-Tree 1.6.12[98,99], allowing for the choice of the most suitable substitution model. Ultrafast bootstrap analysis[100] with 1000 repetitions was used to assess confidence for individual branches. The most likely tree topology was then visualized using the iTOL suite[101], available at https://itol.embl.de.

### Sub-clustering populations with stem cell like properties
To identify markers specific to sub-populations of clusters with stem cell like properties, we subset clusters 0 and 1 (epidermis and coelomic mesoderm) and re-processed these transcriptomes as described above, but without identifying variable features for each sample individually. Newly calculated clusters were then analyzed for their expression of GMP and SAZ related transcripts as well as their UMIs per cell. Strongly positive subpopulations were identified for both clusters. Marker genes for those subpopulations were then calculated against the entire dataset (Supplementary Figs. 4, 5; full marker gene table in Supplementary Data 3).

### Plotting and visualization
All plots displaying single cell data were generated using the R package SCpubr (v. 1.1.2)[102]. The displayed values on gene expression UMAPs are log-transformed transcript counts normalized by UMI.

### in situ Hybridization Chain Reaction (HCR)
For in situ visualization of gene expression, we sampled tissues of either growing or regenerating animals as described above. Labeling was done following our previously published protocol for *Platynereis*[16], with probes designed using the publicly available algorithm HCR3.0 Probe Maker (v. 2021_0.3.2, described in ref. [103]). All HCR probe sequences are available (Supplementary Data 5).

### Whole Mount In Situ Hybridization (WMISH)
The gene sequences for *Platynereis prrx* and *flrtl* were amplified by PCR on cDNA of stage 7 regenerates (7 days post amputation) and cloned into pJet2.1. Primers introducing an Sp6 promotor sequence were used to generate the transcription templates. Digoxigenin-labeled probes were synthesized by Sp6 in vitro transcription, cleaned using the

RNeasy Kit (Quiagen) and stored in hybridisation mix at −20 °C. Nitro blue tetrazolium chloride/5-bromo-4-chloro-3'-indolyphosphate (NBT/BCIP) WMISH was performed as previously described[104]. Samples were permeabilized with proteinase K for 45 s. Bright field pictures for NBT/BCIP WMISH were taken on a Zeiss Z2 Imager, 20x objective.

**Primers to amplify gene sequences.** prrx_for: CGGAATTGCCTCAGCTTACTACTCTC
 prrx_rev: CTGAGCCATCTGGTGGTGGTGG
 flrtl_for: GTTCCCTTGCAGTCACTTT
 flrtl_rev: CACTGTTCCTCTTGCCTTTT

**Primers to generate antisense probe template.** prrx_for: CGGAATTGCCTCAGCTTACTACTCTC
 pJet2.1_sp6_rev: GGATTTAGGTGACACTATAGAACGACTCACTATAGGGAGAGCGGC
 flrtl_for: GTTCCCTTGCAGTCACTTT
 pJet2.1_sp6_rev: GGATTTAGGTGACACTATAGAACGACTCACTATAGGGAGAGCGGC

### Microscopy and image analysis
Confocal images of in situ HCR labelings were taken using a Zeiss LSM700 confocal microscope with a Plan-Apochromat 40x/1.3 Oil DIC, WD 0.21 mm lens, using Zeiss Zen software. Images were then processed using FIJI software[105] for adjustment of contrast, LUT selection and creation of overlay images.

### Quantification of nuclear and nucleolar size in regenerating tissue
To quantify the change in nuclear and nucleolar size of cells responding to posterior injury over time, animals at 0, 12 and 48 hpa were processed and imaged as described above. Nuclei and nucleoli were manually selected as regions of interest (ROIs) in FIJI software by drawing an outline around the areas positive for the nuclear stain (nucleus) and the roughly circular areas within negative for the stain (nucleolus). 50 nuclei and corresponding nucleoli closest to the amputation site were counted per biological replicate ($n = 3$ for each timepoint); nuclei that did not fully span the focal plane were excluded from the analysis. The data were plotted in GraphPad Prism v10.2.2, and the statistical significance of the differences in means between the timepoints was calculated with a one-way ANOVA test with multiple comparisons.

### Molecular cloning
The transgene rps9::H2A::mCherry::F2A::GFP::CAAX (simplified as pHCX) was engineered using the Gibson assembly protocol[106]. The donor plasmids for this construction were pEXPTol2-EGFP-CAAX, pEXPTol2-H2A-mCherry[52] and pTol2{rp-s9::egfp}[15]. The ribosome skip site coding sequence F2A was inserted between the two recombinant fluorescent protein coding sequences by adding it to the cloning primer. For the Gibson reaction, Gibson assembly master mix (NEB, France, E2611S) and NEB 5-alpha Competent E.coli (NEB C2987H) were used, following the manufacturer's protocol.

### Generation and analysis of transgenic animals
*Tol2* transposase mRNA was synthetized using the SP6 mMessage kit from Ambion (AM1340). The manufacturer's protocol was followed until the end of DNase step (1 ml DNase 15 min at 37 °C). For purification of mRNA, MEGAclear kit from Ambion (AM1908) was used, following RNA elution option two with the following modifications: elution buffer was heated to 72 °C, this warmed elution buffer was applied to filter cartridge containing mRNA, the tubes were kept at 72 °C heated plate for 5 min before centrifuging for elution.
 For micro-injections, previous protocols were used[52,107]. Briefly, fertilized eggs were dejellified at 45 min post fertilization, using a

80 mm nylon mesh. The egg cuticle was softened with a flash treatment with proteinase K diluted in seawater (100 mg/ml). After abundant rinsing with sea water, the eggs were micro-injected with a mix of Tol2 transposase mRNA (300 ng/ml) and plasmid pHCX (100 ng/ml). Injected eggs were incubated overnight at 28 °C. Injected batches of larvae and juveniles were raised in a common polypropylene container with 800 ml of sea water until they reached 6 weeks. Worms were then relaxed and amputated as described before. Amputated posterior parts were kept in 7.5% $MgCl_2$ diluted 1:1 with sea water and mounted on slides using three layers of tapes as spacer. Confocal images were acquired on a Zeiss LSM 780 confocal scanning microscope. Amputated animals were dispatched in individual boxes and fed carefully to avoid fouling of the small amount of sea water. The operation was repeated twice after three weeks of regeneration. Some individuals however were not documented for two rounds of regeneration because they underwent sexual maturation, that stops regeneration.

### AZD8055 treatment

Animals were surgically amputated as described above. Regenerating animals were then kept in glass beakers in artificial sea water, either treated with 10 μM AZD8055 (MedChemExpress, USA) or with an equal amount of carrier control (DMSO).

### Reporting summary

Further information on research design is available in the Nature Portfolio Reporting Summary linked to this article.

## Data availability

All primary data generated for this manuscript are available online. The bulk RNA sequencing data generated in this study have been deposited in the Genbank SRA database under accession code NCBI SRA BioProject PRJNA1060927 (SAMN39250368 to SAMN39250382). The single cell RNA sequencing data generated in this study have been deposited in the Genbank SRA database under accession code NCBI SRA BioProject PRJNA1060254 (SAMN39223008 to SAMN39223016). The newly described *Platynereis* genes from this transcriptome have been deposited in the GenBank database, and identifiers are provided in Supplementary Data 2. The processed single cell sequencing data (Seurat object with annotations, metadata and processed reads) is available at NCBI Geo under accession number GSE277281. Confocal imaging stacks for intact and regenerated mosaic transgenic animals are deposited in Figshare (https://doi.org/10.6084/m9.figshare.27046045). Source data are provided with this paper.

## Code availability

Information on single cell data processing is found in the Methods section and in a github repository accessible under the following link: https://github.com/awesomeCells/platynereis_regeneration_SC_atlas_2024.

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

## Acknowledgements

The authors are grateful to members of their respective research groups for continuous input into this project; Alison Cole (University of Vienna) and Simon Haendeler (CIBIV, University of Vienna) for advice on single-cell data processing; Kevin Nzumbi Mutemi and Detlev Arendt (EMBL, Heidelberg) for access to a draft *Platynereis dumerilii* genome sequence and primary annotations; Margaryta Borisova, Andrij Belokurov and Netsanet Getachew (University of Vienna) for animal supply; the Max Perutz Labs BioOptics Facility for access to advanced light microscopy and FACS; the ImagoSeine core facility of Institut Jacques Monod, a member of France BioImaging (ANR-10-INBS-04) and certified IBiSA platform and the Next Generation Sequencing services at the Vienna BioCenter Core Facilities for service and advice. This research has been supported by a joint grant from the Agence Nationale de Recherche (ANR) ANR-16-CE91-0007 (G.B.) and the Austrian Science Funds (FWF) I2972 (F.R.; doi: 10.55776/I2972), the FWF Special Research Project F78 (F.R.; doi:10.55776/F78), the Fondation ARC pour la recherche sur le cancer (grant LSP 190375) (G.B.), the FWF doc.Funds PhD programme DOC 72 "Stem Cells, Tissues, Organoids – Dissecting Regulators of Potency and Pattern Formation (SCORPION)" (F.R.; doi:10.55776/

DOC72), a post-doctoral fellowship of the Laboratory of Excellence (LABEX) "Who am I?" ANR-11-LABX-0071 (B.D.O.), the Austrian Academy of Sciences (ÖAW) DOC fellowship programme (A.W.S.), the German Academic Scholarship Foundation (L.A.), and the University of Vienna Research Platform "Single Cell Regulation of Stem Cells (SinCeReSt)" (F.R.). For open access purposes, F.R. has applied a CC BY public copyright license to any author accepted manuscript version arising from this submission.

## Author contributions

Conceptualization: A.W.S., L.A., G.B., F.R. Investigation: A.W.S., L.A., M.F., C.R., N.M., G.B., F.R. Data curation: A.W.S., M.F., G.B. Methodology: A.W.S., L.A., M.F., B.D.O., G.B., F.R. Funding acquisition: A.W.S., L.A., B.D.O., G.B., F.R. Supervision: G.B., F.R. Visualization: A.W.S., L.A., N.M., G.B., F.R. Writing: A.W.S., L.A., G.B., F.R.

## Competing interests

The authors declare no competing interests.
