## [Peer Review file · Nature Communications]

Molecular profiles, sources and lineage restrictions of stem cells in an annelid regeneration model

Corresponding Author: Dr Florian Raible

Version 0:

Reviewer comments:

Reviewer #1

(Remarks to the Author)

In "Molecular profile, source and lineage restriction of stem cells in an annelid regeneration model", Stockinger and co-authors provide an unprecedented and impressive dataset that characterises posterior regeneration in the model annelid *Platynereis dumerilii* at single-cell resolution and with transgenics. Animal regeneration (particularly whole-body regeneration) is a fascinating capability that is, however, still poorly understood in most animal groups. Annelids (i.e. segmented worms) show great regenerative capacities, with anterior (head) and posterior (tail) regeneration being likely ancestral traits to the entire clade and even some lineages relying almost entirely on whole-body regeneration as an (asexual) reproductive strategy. The annelid *Platynereis dumerilii* has been established as a model to investigate animal development and regeneration in the last two decades. *Platynereis* can regenerate the posterior end after amputation, driven by the formation of a blastema and differentiation of the missing cell types and body parts. In *Platynereis*, and likely in most other annelids, the dedifferentiation of pre-existing cells in the pre-blastema region has been hypothesised to be the source of blastema cells. However, to what extent these dedifferentiated cells remain lineage committed and how dedifferentiation/regeneration unfolds over time at a cellular level is unclear. The authors set to respond to these fundamental questions employing single-cell and transgenic approaches. They perform a replicated time course of single-cell transcriptomic profiling of four (+1 additional unreplicated point at 12 hpa) points from amputation to full regeneration of the posterior tip of the animal. This unprecedented dataset for an annelid corroborated many of the previously known expression dynamics during regeneration and suggested the existence of lineage-restricted dedifferentiated progenitor cells. The authors then used mosaic transgenesis to generate animals for which broad adult tissues (e.g., epidermis, muscles, central nervous system, gut) were labelled and used those to identify how those cell types appeared anew during regeneration, which supported the existence of lineage-restricted regeneration (e.g., the muscles for muscles, the epidermis form epidermis, and not vice versa). Finally, the authors test the role of the TOR pathway during regeneration, demonstrating that normal TOR activity is required for successful regeneration. Altogether, this is an important technical advance in the study of annelid regeneration and an incredible resource that will benefit and be broadly used by the community. It convincingly supports previous scenarios about the cellular and molecular mechanisms driving annelid regeneration and indicates that, as in some vertebrates, dedifferentiation might be an essential mechanism for whole-body regeneration and tissue repair.

There are, however, several important points that the authors might want to consider and would solidify their work:

- Candidate gene approaches largely drive the authors' extensive single-cell dataset analyses. That might be a good approach to validate that the single-cell transcriptomic dataset recapitulates the expression and dynamics of genes already known to be involved in posterior regeneration. However, this dataset can offer much more, and the authors have under-explored it. What are the new genes and genome-wide dynamics revealed by this time course of single-cell transcriptomic profiling? For example, cluster number 16 (midgut) appears to be almost exclusive of the 0 hpa, whereas cluster 8 (smooth muscle) is nearly non-existent at that time point (Supp Fig 2d). What does that mean for our understanding of posterior regeneration in *Platynereis*? A replicated time course also allows the identification of differentially expressed genes in pairwise comparisons and computationally inferring transcriptional trajectories between cells and time points. These are all commonly performed and well-documented analyses in single-cell transcriptomic studies that would maximise the knowledge gained through the authors' impressive dataset and qualify or further strengthen their conclusions. Likewise, the authors highlight on multiple occasions that their dataset will allow cross-species comparisons and identify similarities/differences in the regeneration process. Although a full comparison is out of the scope of this work, some steps in

that direction are analytically feasible and would be very informative.

- The narrative of the paper could also be improved. It seems three different analyses (single-cell transcriptomics, transgenesis and TOR analyses) were independently made and then combined to make a story. Are there unbiased, genome-wide analyses of the single-cell transcriptomic data that suggest the role of the TOR pathway in regeneration (e.g., broadly unregulated at some point in the process)? Likewise, the reconstruction of transcriptional trajectories in the single-cell data could support the observations with transgenic lines and tie all the different experimental approaches (single-cell transcriptomics, transgenesis, and functional perturbations) and prior knowledge of candidate genes (e.g., up regulation of *myc*) more coherently in a single story.

- The HCRs to co-detect *Hox3* expression with putative subpopulation-specific stem cell markers are somewhat unclear, especially for *ntr* and *p2x* at stage 3, where it is difficult to discern the true signal from speckles and noise. It might help if the authors provide a regular NBT/BCIP in situ or an HCR of the general expression pattern for these genes, which will give a broad idea of how these markers compare to *Hox3* expression before going to the fine detail of near sub-cellular coexpression.

- The authors use the terminology "germ layers" to refer to the tissues and cell types of the adult. However, germ layers apply to the embryo during gastrulation, and their correspondence to adult tissues is often unclear (e.g., muscles might come from different germ layers, etc). I suggest using terms like epidermis, muscle, gut, etc, to refer to tissues instead of ectoderm, mesoderm and endoderm. In fact, the authors' data show one such case with the ectodermal transgenic, which initially does not cover the pygidia ectoderm (as they have different developmental origins), but after regeneration, the pygidium ectoderm is formed from the pre-existing trunk ectoderm. Some sentences are also unclear. E.g., "While these transgenic clones do not demonstrate the embryonic germ layer origins, they show that germ layers remain strictly compartmentalized during posterior segment addition, similar to embryonic/larval development". What do the authors mean by "embryonic germ layer origins"? Leaving the germ layer paradigm and focusing on the adult tissues will make the story more accessible.

- Figure 3 is a bit dense and could be split into two (e.g., a description of the transgenics and the study of the contribution during regeneration). Likewise, a summary/conclusion figure would help communicate the main message of the manuscript and highlight the advancement and outstanding questions.

Other more minor comments:

- "Due to the evolutionary placement of annelids in the lophotrochozoan superphylum" is a vague statement. Annelids are as well positioned as any other lophotrochozoan/spiralian for long-range comparisons.

- Supp Fig 1A and Fig1A are the same.

- How well do scRNAseq datasets compare with the bulk RNAseq initially generated? The bulk RNAseq dataset is under-used in the manuscript.

- Seurat starts cluster numbering from 0, but it might be easier to renumber them starting with cluster 1?

- Supp Fig 2d, it might be helpful to show how these proportions change between replicates (at least for 0, 24, 48 and 72hpa)

- Figure 3: add a yellow arrow to panel j?

- Line 635: "We normalized" (change capital)

- Line 809: should be (u-z) in the caption.

Reviewer #2

(Remarks to the Author)

Stockinger et al. sought to characterize the stem cells that mediate the regeneration of the new posterior tissues in *Platynereis dumerilii* upon amputation. By employing single-cell RNA sequencing and further functional exploration of wound-adjacent cell populations, authors identified putative posterior stem cells (PSCs) that emerge in the blastema of the sampled fragments, potentially serving as the source of these new posterior tissues. In addition, by establishing a Tol2-mediated transgenesis protocol in *Platynereis*, using a construct that contains both a nuclear mCherry and a membrane EGFP fluorescent proteins, under the control of the ubiquitous *rps9* promoter, authors traced the regeneration of transgenic individuals and inferred that germ layers are compartmentalized— each germ layer will regenerate tissues of the same germ layer identity. Finally, as in other systems, TOR signaling is involved in the response to wounding and subsequent regeneration in *Platynereis*. Altogether, this study proposes that upon amputation, *Platynereis* requires TOR signaling to employ germ layer-specific PSCs to re-establish lost tissues during posterior regeneration.

This is a valuable study, as it advances the resources and protocols available for *Platynereis*, specifically with the new single-cell transcriptomic atlas for posterior regeneration and the successful Tol2-mediated transgenesis protocol. While the data reported in the manuscript are clear, clarifications of their inferences, as well as further experimentation and quantifications, will strengthen the article. Below are the details on these major comments.

1) Provide alternative possibilities that could be underlying regeneration in *Platynereis*

The introduction of the manuscript highlights that existing data elsewhere suggest that some type of dedifferentiation mechanism (namely re-embryonalization) mediates posterior regeneration in *Platynereis*. While lines 419-421 and 515-517 provide an alternative possibility to dedifferentiation, invoking progenitor cells that are specific to a given cell lineage or germ layer, dormant or quiescent stem cells, lineage-specific progenitors etc., these should be highlighted across the entirety of the article. The inferences made during the results section are exclusively focused on a hypothesis that relies on a dedifferentiation model— authors should outline alternative hypotheses that could explain the data. Hypertranscription, cellular morphologies at a given time point during regeneration (e.g., enlarged nucleoli), and the expression of stem cell- and

proliferation-related genes might be suggestive of stem cells, but they are not evidence solely for dedifferentiation. See below examples of lines in the manuscript where authors focus on a dedifferentiation hypothesis and provide no alternative hypotheses:

- Line 229: "hinting at sources of dedifferentiated PSCs"
- Lines 243-244: "consistent with multiple, parallel de-differentiation events occurring in distinct populations of wound-adjacent cells"
- Line 268: "Having found evidence for distinct pools of dedifferentiated progenitors"
- Lines 311-312: "distinct populations of mesodermal and ectodermal PSCs derive from existing cells by dedifferentiation"

2) Provide intermediate cellular states during regeneration using the transgenic lines

By means of Tol2-mediated mosaic transgenesis and regeneration assays, authors claim that regeneration of new tissues will come from cells of the same cellular identity (e.g., a mesodermal clone will regenerate cells of the mesoderm). This is true if cells of that lineage, which have inserted the transgenic construct, are undergoing differentiation— an alternative explanation would be that the transgenic construct was inserted in a genomic loci that falls under the control of a lineage-specific enhancer. In the latter possibility, the transgene is expressed whenever that enhancer is active, which could happen in uninjured animals, as well as in the blastema and later regenerated tissues, in a seemingly "restricted cellular lineage". In order to rule out that the transgene is getting expressed because it is limited to the given cellular lineage rather than by the control of an enhancer, authors should provide figures highlighting transgenic cells that are undergoing this differentiation process. While authors show in Figure 3 the start and end point of this regeneration process, it is important show the intermediate states of those cells that are differentiating.

3) Experiments and quantifications to strengthen authors' claims

Authors provide evidence showing that populations of cells adjacent to the wound, inferred to be the ectodermal and mesodermal PSCs, are expressing top markers of their respective clusters (cluster 0 and cluster 1, respectively). Based on the computational analysis of the single-cell RNA-seq datasets, these two major clusters show high expression of GMP-related genes and genes associated with proliferation, leading authors to claim that these cells adjacent to the wound "express stem cell-related genes and also enter the cell cycle upon injury" (lines 104-106), while also claiming that there is an "absence of obvious proliferation" (lines 518-519) during the sampled regeneration time points. Authors should clarify those statements and also provide functional data that shows that these wound-adjacent cells are expressing the presented GMP-related genes (*piwi*, *vasa*, *nanos*), and that they are proliferative during regeneration (or alternatively show that there is no obvious proliferation response).

In addition, images in Figure 2 are shown to highlight the cells with enlarged nucleoli, suggestive of stem cells, in stages 1 and 3. It would be helpful to establish a definition of "enlarged nucleoli" (how to distinguish such a nucleolus from other nucleolus) and provide quantification of those cells with enlarged nucleoli across the regeneration time points, starting at stage zero, to assess whether there is a change of the number of cells with this morphology as regeneration progresses.

Minor comments:

- Lines 205-207 and Figure 1e: Why is the posterior gut marker considered to be 're-emerged' at 12-72hpa? Wouldn't the 0hpa time point samples also contain *foxA*⁺ cells in the gut? Also, there is a strong expression of *foxA* in neuronal cells that doesn't seem to be mentioned in the text.
- While seeing the regeneration of new tissues in Figure 3H, it is harder to tell where regeneration is happening in Figures 3I and 3J. Perhaps adding markers or arranging the panels will make it easier to identify the regeneration of these new tissues.

Reviewer #3

(Remarks to the Author)

The manuscript provided by Raible, Balavoine, Özpolat and colleagues is a formidable example for a study significantly advancing our understanding in a field of general interest such as posterior regeneration, in a non-model species that is especially relevant for this field. The regeneration of the posterior body axis occurs in vertebrates such as axolotl; yet, single-cell resolution has not yet been achieved for this process. In this study, the authors have chosen the polychaete *Platynereis dumerilii*, which has already been introduced as a model for posterior regeneration. Instead of a totipotent blastema they identify distinct stem cells for ectodermal and mesodermal lineages, forming from the epidermis and from the coelomic epithelium, respectively. Furthermore, taking advantage of the generation of transient transgenic F0 animals showing clonal inheritance of a ubiquitously expressed transgenes, they provide evidence for strict lineage restriction during regenerative growth which traces back to developmental compartments. The study thus provides a first single-cell resolution framework for an important experimental paradigm in posterior regeneration and should thus spark considerable interest of an extended readership. I have some more relevant and few minor comments that the authors should address before publication.

1) What happens to the gut? The authors find convincing upregulation of specifying transcription factors such as *cdx* and *FoxA* in clusters 11 and 20 which represent the gut but then they describe stem cells only for epidermis-derived ectodermal cells and coelomic epithelium-derived mesodermal cells. How do endodermal mid- and hindgut cells form? The authors speculate that endodermal stem cells "are spread in a diffuse way along the length of the trunk". Can this be substantiated by *cdx* and *foxa* expression? Given the continuity of coelomic and gut epithelium – was there ever a co-labelling of the two tissue types?

2) The authors introduce the concept of hypertranscriptomic stem cells recognized by high UMI count. Now, high UMI counts can also be accounted for by doublet formation. Have the authors controlled for doublets?

3) How about additional segmented structures as chaetae and nephridia: Did the authors find evidence for new proliferative lineages also for these structures?

4) What are the extra nuclei labelled in the enodermal clone of 3e, spreading out in a bilaterally symmetrical pattern into the pygidium?

5) In their very insightful discussion the authors discuss similarities in the roles of *msx* and *sp9* in axolotl, as well as of *prx* in chicken and mouse limb development. How does their new paradigm relate to posterior axis regeneration as occurs in salamander?

In summary, the authors are to be congratulated for an insightful and thought-provoking study that will significantly advance the field of axis regeneration.

Version 1:

Reviewer comments:

Reviewer #1

(Remarks to the Author)

The authors have consistently and adequately addressed all my previous suggestions. This is a significant piece of work and an important resource that will broadly benefit the community.

Reviewer #2

(Remarks to the Author)

Authors have revised and implemented all of the points that were raised carefully:

A) Authors have expanded on the possible sources of cells that underly the regeneration in *Platynereis* in the introduction of the manuscript, by adding citations to existing literature that address this question, and built a strong case arguing how dedifferentiation is likely the mechanism that allows posterior regeneration to happen. It is now clear how previous work suggests dedifferentiation as the most likely mechanism and why the authors interpret their results with this hypothesis as the primary one.

B) In our initial round of revision, we had asked authors to provide detailed imaging and quantification of cells adjacent to the wound sites to support their hypothesis model of regeneration, which is now specifically focused on the dedifferentiation hypothesis. New data provided by the authors suggest that cells of a differentiated type undergo changes indicative of dedifferentiation upon injury: 1) the expression of key markers of their respective differentiated identity are decreased upon injury (e.g., loss of the epidermal-specific *col6a6* in cells adjacent to the wound site), 2) significant increase of the area occupied by the nucleus and nucleolus of these cells that are immediate to the wound site, a morphological property that is reminiscent of the high-potency Annelid teloblasts, and 3) the *de novo* expression of the GMP marker *piwi* and incorporation of EdU in these cells, indicative of proliferative cells that are otherwise absent prior to the injury.

C) Finally, authors carefully addressed our major concern about the possibility of Tol2-mediated genomic insertion in loci under the control of unexpected lineage-specific enhancers. By increasing both the sampling number of transgenic individuals and the repeated and independent identification and imaging of labeled cells in the expected body plan domains, authors have addressed our concerns by making sure that their inferences using these transgenic tools are unlikely to be a consequence of enhancer trapping but rather a biological phenomena suggestive of dedifferentiation.

Remaining minor comments:

- Sentence in lines 92-94 should be improved for clarity
- Typo in line 290: double "ii"
- Typo in line 457: double "and and"

Reviewer #4

(Remarks to the Author)

Stockinger, Adelman et al. :: Point-to-point responses to the reviewers

Reviewer #1 (Remarks to the Author):

In "Molecular profile, source and lineage restriction of stem cells in an annelid regeneration model", Stockinger and co-authors provide an unprecedented and impressive dataset that characterises posterior regeneration in the model annelid *Platynereis dumerilii* at single-cell resolution and with transgenics. Animal regeneration (particularly whole-body regeneration) is a fascinating capability that is, however, still poorly understood in most animal groups. Annelids (i.e. segmented worms) show great regenerative capacities, with anterior (head) and posterior (tail) regeneration being likely ancestral traits to the entire clade and even some lineages relying almost entirely on whole-body regeneration as an (asexual) reproductive strategy. The annelid *Platynereis dumerilii* has been established as a model to investigate animal development and regeneration in the last two decades. *Platynereis* can regenerate the posterior end after amputation, driven by the formation of a blastema and differentiation of the missing cell types and body parts. In *Platynereis*, and likely in most other annelids, the dedifferentiation of pre-existing cells in the pre-blastema region has been hypothesised to be the source of blastema cells. However, to what extent these dedifferentiated cells remain lineage committed and how dedifferentiation/regeneration unfolds over time at a cellular level is unclear. The authors set to respond to these fundamental questions employing single-cell and transgenic approaches. They perform a replicated time course of single-cell transcriptomic profiling of four (+1 additional unreplicated point at 12 hpa) points from amputation to full regeneration of the posterior tip of the animal. This unprecedented dataset for an annelid corroborated many of the previously known expression dynamics during regeneration and suggested the existence of lineage-restricted dedifferentiated progenitor cells. The authors then used mosaic transgenesis to generate animals for which broad adult tissues (e.g., epidermis, muscles, central nervous system, gut) were labelled and used those to identify how those cell types appeared anew during regeneration, which supported the existence of lineage-restricted regeneration (e.g., the muscles for muscles, the epidermis form epidermis, and not vice versa). Finally, the authors test the role of the TOR pathway during regeneration, demonstrating that normal TOR activity is required for successful regeneration. Altogether, this is an important technical advance in the study of annelid regeneration and an incredible resource that will benefit and be broadly used by the community. It convincingly supports previous scenarios about the cellular and molecular mechanisms driving annelid regeneration and indicates that, as in some vertebrates, dedifferentiation might be an essential mechanism for whole-body regeneration and tissue repair.

We thank the reviewer for their overall very positive remarks on our manuscript. We hope that the additional analyses and resources that we have included in the revised version in response to the review process help to clarify the outstanding aspects and will make it even easier to use the data for the interested research community.

There are, however, several important points that the authors might want to consider and would solidify their work:

- Candidate gene approaches largely drive the authors' extensive single-cell dataset analyses. That might be a good approach to validate that the single-cell transcriptomic dataset recapitulates the expression and dynamics of genes already known to be involved in posterior regeneration. However, this dataset can offer much more, and the authors have under-explored it. What are the new genes and genome-wide dynamics revealed by this time course of single-cell transcriptomic profiling?

We thank the reviewer for suggesting the inclusion of additional systematic analyses.

In the revised version, we have now included several systematic analyses that were not part of the original manuscript:

A CytoTRACE analysis has helped us to delineate possible transcriptional dynamics occurring within every cell population as it responds to injury. Among other benefits, this approach helped to more precisely delineate dynamic transcriptional changes accompanying the re-emergence of stem cell signatures described in our manuscript.

Likewise, we have performed a Gene Ontology (GO) analysis that now provides GO terms associated with the transcriptional changes within each cluster. These data are presented in the revised Fig. 1i and have been uploaded as Supplementary Data 4.

Not only do our revised data provide a more comprehensive resource that is more easily accessible to the readership of the study, but these data can also be combined with other aspects of our study. For instance, combining the time information from the sampling scheme with the CytoTRACE analysis, we could reveal the directionality of transcriptional changes, in additional support of the de-differentiation concept we propose for clusters #0 and #1. To further test this concept, we have capitalized on the identification of new differentiation markers to design a more precise experiment testing the gradual loss of differentiation in the epidermal cell population (s.b.). By making these data available along with the manuscripts, readers can use these more comprehensive resources to trace changes in bulk RNA transcriptomics to cell clusters responsible for the expression of the respective genes.

For example, cluster number 16 (midgut) appears to be almost exclusive of the 0 hpa, whereas cluster 8 (smooth muscle) is nearly non-existent at that time point (Supp Fig 2d). What does that mean for our understanding of posterior regeneration in *Platynereis*?

We thank the reviewer for pointing out this apparent discontinuum and allowing us to clarify those cases in which clusters are seemingly restricted to specific time points of the sampling.

In our revised section on “Molecular repatterning and emerging stem cell-like properties in distinct cell populations”, we now more clearly explain that in two cases (clusters 16/4 ; clusters 14/8), the clustering algorithm has assigned distinct clusters to cell populations obtained from the freshly amputated animals, while the adjacent cluster contains the corresponding cells from the later sampling time points. As we explain in the revised text, this split of the cell populations likely reflects significant changes in gene expression in response to injury in gut and smooth muscle cell populations, respectively. In both tissues, this is consistent with morphallactic repatterning.

As the reviewer’s comment made us aware of, the split in distinct clusters was not explained clearly enough in the original version, and should now be more understandable.

A replicated time course also allows the identification of differentially expressed genes in pairwise comparisons and computationally inferring transcriptional trajectories between cells and time points. These are all commonly performed and well-documented analyses in single-cell transcriptomic studies that would maximise the knowledge gained through the authors’ impressive dataset and qualify or further strengthen their conclusions.

As outlined above, we have included systematic CytoTRACE analyses for the different cell populations. We have also used these to map genes whose expression correlates with the respective CytoTRACE scores. We consider the dynamicity of the transcriptome data one of the central and biologically interesting aspects of our model system and study, which is why we have opted for this correlation approach rather than pairwise comparisons.

As we show in revised Fig. 2f, this correlation analysis has predicted a highly useful marker (*col6a6*) that we could use to validate the gradual loss of epidermal cell identity accompanying the emergence of cytological stem cell features within the same cell population (cf. Fig. 2g). This strengthens our interpretation that the new stem cells emerge by de-differentiation from differentiated tissues. The systematic data have been made available in the revised Supplementary Data 4, providing a rich resource for future analyses by anyone interested.

Likewise, the authors highlight on multiple occasions that their dataset will allow cross-species comparisons and identify similarities/differences in the regeneration process. Although a full comparison is out of the scope of this work, some steps in that direction are analytically feasible and would be very informative.

We share the reviewer's interest in a more systematic cross-comparison of regeneration data across different model systems, but also feel that this is a non-trivial task that merits a separate analysis. For instance, we are aware of single-cell RNAseq data in tail regeneration-competent and -incompetent tadpoles of the frog *Xenopus laevis* (Aztekin et al., Science 364, 653–658 (2019)) that has proposed a role for specific epidermal cells in the regeneration process. While this might seem an interesting starting point for comparing the epidermal cells across these systems, the tadpole is still a developmental state, and the role of the epidermis as a signaling center in *Xenopus* tail regeneration seems to differ significantly from the role that we assign to epidermal cells as source for the re-emergence of stem cells in the worm. In turn, whereas we see an interesting perspective for the comparison of dedifferentiation processes in the axolotl system, axolotl tail regeneration has so far not yet been analysed on the single-cell level, preventing a more direct comparison between cell populations, which of course may nonetheless be very challenging across this large evolutionary distance.

The phylogenetically closest system that also seems very attractive for comparisons is the single-cell dataset for *Pristina leydi* that has now been published by the group of Jordi Solana (Álvarez-Campos et al. Nat Commun 15, 3194 (2024)). Even though this study features stem cells in another annelid model,, the two studies are far more complementary than comparable, because the Álvarez-Campos study focuses on the occurrence and potency of stem cells in a homeostatic state, excluding the processes that occur during regeneration or paratomic fission of these systems.

Nonetheless, we have now included a section in the discussion in which we discuss the commonalities and differences between the Álvarez-Campos study in *Pristina leydi*. We anticipate that this comparison will be an interesting subject for a more dedicated scientific discussion of the shared and divergent roles of stem cells in annelids. Also, we anticipate that our study in *Platynereis* will provide a valuable template for any future resources interrogating regeneration or fission in *Pristina*.

- The narrative of the paper could also be improved. It seems three different analyses (single-cell transcriptomics, transgenesis and TOR analyses) were independently made and then combined to make a story. Are there unbiased, genome-wide analyses of the single-cell transcriptomic data that suggest the role of the TOR pathway in regeneration (e.g., broadly unregulated at some point in the process)?

Likewise, the reconstruction of transcriptional trajectories in the single-cell data could support the observations with transgenic lines and tie all the different experimental approaches (single-cell transcriptomics, transgenesis, and functional perturbations) and prior knowledge of candidate genes (e.g., up regulation of myc) more coherently in a single story.

We thank the reviewer for the suggestions to better connect the different experimental approaches, and we have followed their advice.

Generally, we aimed at increasing cross-references between the different parts of the manuscript. In keeping with this, we also introduced a new item in the transition between the bulk RNA sequencing and the single-cell work, where we show that the temporal progression in the single-cell dataset matches with the temporal progression in the bulk data (Supplementary Fig 1c).

Moreover, we made an effort to validate the computationally predicted dedifferentiation process using an experimental approach. Specifically, we have used the CytoTRACE analysis that we have performed (s.a.) to identify a marker (*collagen 6a / col6a*) that was predicted to be present in differentiated epidermis cells and become gradually downregulated. As we now show in the context of the revised Figure 2, this marker is indeed downregulated while the tissue exhibits a quick upregulation of nucleolar and nuclear size (s.b.).

Likewise, and in direct response to the reviewer's suggestion, we now refer to the newly included systematic GO-term analysis to provide additional evidence that indeed, TOR signaling is suggested to be involved in the early regenerative response, an element that was indeed missing from the initial manuscript. In addition to TOR itself, we also include data on all components of the "Ragulator" complex that support the very same notion. Besides more literature-based arguments that TOR acts as a regulator of stemness and proliferation in other systems, we think that this now provides a stronger biological motivation for the successful validation experiments included in the study.

Concerning the reviewer's suggestion for better linking the transgenic work (and the specific request to present differentiation trajectories to support the lineage restriction data), we would like to clarify that the single-cell dataset was designed to capture the initial stages of regeneration up to the re-establishment of stem cells, but not the stage where these stem cells give rise to differentiated tissue. Consistent with this, our main focus for the scRNAseq analysis is on the re-emergence of these *bona fide* stem cells, predicting that there are distinct populations emerging from the epidermis (a tissue of ectodermal origin) and the subepidermal layer (a tissue of mesodermal origin). This was a significant finding, as distinct paths for the re-emergence of posterior stem cells had been hypothesized about, but not demonstrated on the molecular level. Due to the focus of the time course on early regenerative events, the scRNAseq dataset does not (yet) show a connection between the respective *bona fide* stem cells and newly formed differentiated tissue that could be used to calculate differentiation trajectories as suggested by the reviewer. Such connections would only be expected to become visible at later stages not covered in the present analysis. Instead, the transcriptomic changes underlying DEdifferentiation were captured and are presented in the manuscript. The newly added CytoTRACE analysis makes these dynamics more easily accessible for future studies.

In turn, the transgenic data allowed us – in a complementary fashion – to interrogate possible restrictions in differentiation, as we could trace the offspring of transgenically labeled cells. As we are bound to the broad developmental domains labeled by the approach, we cannot trace the offspring of individual cells – and hence remain necessarily blind concerning how many subtypes each *bona fide* stem cell population might harbour. Nonetheless, our data are very clear with respect to the overall restriction that cells originating from labeled cells of ectodermal origin do not contribute to tissues developmentally formed from meso- or endoderm, and *vice versa*. Again, this finding is significant, as it differs from a scenario expected for pluripotent cells as they exist in planarians, and have recently also been suggested to underlie tissue homeostasis in the annelid *Pristina leydi*. This aspect is now also briefly discussed in the discussion section of the revised manuscript.

Taken together, we believe that these measures have produced a more coherent and consistent narrative, as suggested by the reviewer.

- The HCRs to co-detect Hox3 expression with putative subpopulation-specific stem cell markers are somewhat unclear, especially for *ntr* and *p2x* at stage 3, where it is difficult to discern the true signal from speckles and noise. It might help if the authors provide a regular NBT/BCIP in situ or an HCR of the general expression pattern for these genes, which will give a broad idea of how these markers compare to Hox3 expression before going to the fine detail of near sub-cellular coexpression.

Following the reviewer's suggestion, we now provide also traditional whole-mount *in situ* hybridisation data for two representative stem cell markers – *ntr* and *prrx* – in comparison to *hox3* expression. These additions should allow the reader to more easily comprehend the overall relationship of spatial domains, and also support the different time points at which the *bona fide* stem cells become molecularly discernible. The respective images are provided in Supplementary figure 4e. We also completely revised Figure 2 and performed new stainings in response to other reviewer suggestions, and provide new and clearer images of gene expression in Figure 2h-k.

- The authors use the terminology "germ layers" to refer to the tissues and cell types of the adult. However, germ layers apply to the embryo during gastrulation, and their correspondence to adult tissues is often unclear (e.g., muscles might come from different germ layers, etc). I suggest using terms like epidermis, muscle, gut, etc, to refer to tissues instead of ectoderm, mesoderm and endoderm. In fact, the authors' data show one such case with the ectodermal transgenic, which initially does not cover the pygidia ectoderm (as they have different developmental origins), but after regeneration, the pygidium ectoderm is formed from the pre-existing trunk ectoderm. Some sentences are also unclear. E.g., "While these transgenic clones do not demonstrate the embryonic germ layer origins, they show that germ layers remain strictly compartmentalized during posterior segment addition, similar to embryonic/larval development". What do the authors mean by "embryonic germ layer origins"? Leaving the germ layer paradigm and focusing on the adult tissues will make the story more accessible.

We thank the reviewer for requesting a clearer distinction between developmental origin and later tissue restriction. This is an important difference, and indeed, the terminology used in the original manuscript was not precise enough to make this distinction transparent. However, we also would like to point out that the terms “ectodermal” and “mesodermal” have been used in conjunction with regenerated stem cells and their offspring in the segment addition zone by other researchers in the field (see e.g. Gazave et al. *Dev Biol* 382:246–267. doi:10.1016/j.ydbio.2013.07.013; Planques et al. *Dev Biol* 445:189–210. doi:10.1016/j.ydbio.2018.11.004; Boilly et al. *Dev Genes Evol* 1–11. doi:10.1007/s00427-024-00713-5).

In order to remain faithful to preceding literature and yet acknowledge the distinction raised by the reviewer, we have now used a clearer terminology throughout the revised manuscript, including the following changes:

- In the Introduction, the last paragraph has been changed from “separate pools of germ-layer PSCs [...] regenerated from cells of the corresponding germ layers.” to “separate pools of lineage-restricted PSCs [...] regenerated from cells originating from distinct embryonic germ layers.”
 - When first addressing this issue in the results, we now clarify the used terminology: “we first investigated the expression of the homeobox gene *hox3*, whose transcripts are rapidly upregulated in posterior regeneration of *Platynereis dumerilii* (Pfeifer et al., 2012) and mostly restricted to a population of PSCs that are generally referred to as ectodermal PSCs in accordance with their presumed developmental origin (Gazave et al., 2013; Planques et al., 2019).”
 - The section termed “Ecto- and mesodermal PSCs exhibit shared and distinct molecular signatures” has been renamed into “PSCs of ecto- and mesodermal origin exhibit shared and distinct molecular signatures”
 - Accordingly, at various places, the terms “ectodermal PSCs” and “mesodermal PSCs” have been replaced by “ectoderm-derived PSCs” and “mesoderm-derived PSCs”, respectively
 - In the section “Clonal analysis by mosaic transgenesis reveals germ-layer based lineage restriction of posterior growth and regeneration” and Fig. 3, we replaced “germ layers” by “clonal lineages” and “developmental compartments”, respectively
 - In the discussion, the clonal results are summarized as “*Platynereis* regeneration relies on cells with limited potency that respect the distinction between cells arising from different germ layers in development.”
- Figure 3 is a bit dense and could be split into two (e.g., a description of the transgenics and the study of the contribution during regeneration). Likewise, a summary/conclusion figure would help communicate the main message of the manuscript and highlight the advancement and outstanding questions.

We thank the reviewer for suggesting these changes. In line with the suggestion, the original Fig. 3 has now been split into two Figures (Fig. 3, Fig. 4). Concerning the suggestion to provide a graphical summary, due to space constraints, we have not provided a single summary figure, but have made sure that when revising the figures, we include schemes, overview images or labels for clarifying the respective messages (e.g. Fig.2 f, Fig. 3f-i, Fig. 4e,f).

Other more minor comments:

- "Due to the evolutionary placement of annelids in the lophotrochozoan superphylum" is a vague statement. Annelids are as well positioned as any other lophotrochozoan/spiralian for long-range comparisons.

Rephrased accordingly

- Supp Fig 1A and Fig1A are the same.

While these schemes appear very similar, time series do differ (12hpa were not sampled for bulk sequencing; 96hpa were not sampled in single cell sequencing). This difference and the fact that the 12hpa time point was introduced to increase temporal resolution at a critical time window is now clearly stated in the revised results.

- How well do scRNAseq datasets compare with the bulk RNAseq initially generated? The bulk RNAseq dataset is under-used in the manuscript.

The bulk RNA sequencing was used to detect the global gene expression dynamics and refine the timeline for sampling in single cells, which is now more explicitly stated (see point above).

In order to improve the coherence between the bulk RNA and scRNAseq analyses, we now also introduced a new item in the transition between the bulk RNA sequencing and the single-cell work, where we show that the temporal progression in the single-cell dataset matches with the temporal progression in the bulk data (Supplementary Fig 1c).

- Seurat starts cluster numbering from 0, but it might be easier to renumber them starting with cluster 1?

While we agree that the 0-indexed nomenclature the Seurat analysis pipeline uses by default can be unintuitive, it has ultimately been established in the field and used in many publications. Such cases can even be found in previous single cell work on *Platynereis* ([https://www.cell.com/cell/pdf/S0092-8674\(21\)00876-X.pdf](https://www.cell.com/cell/pdf/S0092-8674(21)00876-X.pdf)), in planarians (<https://www.nature.com/articles/s41467-023-39016-0>) and others. We therefore consider it more consistent, reproducible and less prone to misunderstandings to maintain the established default parameters for cluster numbering.

- Supp Fig 2d, it might be helpful to show how these proportions change between replicates (at least for 0, 24, 48 and 72hpa)

We thank the reviewer for pointing this out and added a panel showing the relative contribution to each cluster per biological replicate (Supplementary Fig. 2d), in addition to the previous panel showing contribution per sampling time point (now Supplementary Fig. 2e).

Supplementary Data 1 (sheet 2) further shows the exact numbers of cells in each cluster, split by both biological replicate and sampling time point.

- Figure 3: add a yellow arrow to panel j?

As pointed out in the legend, arrows were meant to point at rings of putative posterior stem cells in the respective focal plane. This is not applicable to the endodermal image, hence the lack of an arrow was intended. The legend now refers more explicitly to the focal plane.

- Line 635: "We normalized" (change capital)

Done

- Line 809: should be (u-z) in the caption.

Done

Reviewer #2 (Remarks to the Author):

Stockinger et al. sought to characterize the stem cells that mediate the regeneration of the new posterior tissues in *Platynereis dumerilii* upon amputation. By employing single-cell RNA sequencing and further functional exploration of wound-adjacent cell populations, authors identified putative posterior stem cells (PSCs) that emerge in the blastema of the sampled fragments, potentially serving as the source of these new posterior tissues. In addition, by establishing a Tol2-mediated transgenesis protocol in *Platynereis*, using a construct that contains both a nuclear mCherry and a membrane EGFP fluorescent proteins, under the control of the ubiquitous *rps9* promoter), authors traced the regeneration of transgenic individuals and inferred that germ layers are compartmentalized— each germ layer will regenerate tissues of the same germ layer identity. Finally, as in other systems, TOR signaling is involved in the response to wounding and subsequent regeneration in *Platynereis*. Altogether, this study proposes that upon amputation, *Platynereis* requires TOR signaling to employ germ layer-specific PSCs to re-establish lost tissues during posterior regeneration.

This is a valuable study, as it advances the resources and protocols available for *Platynereis*, specifically with the new single-cell transcriptomic atlas for posterior regeneration and the successful Tol2-mediated transgenesis protocol. While the data reported in the manuscript are clear, clarifications of their inferences, as well as further experimentation and quantifications, will strengthen the article. Below are the details on these major comments.

We thank the reviewer for their overall very positive comments on our work, and for the constructive suggestions that helped us to strengthen the conceptual points of the revised manuscript.

1) Provide alternative possibilities that could be underlying regeneration in *Platynereis*
The introduction of the manuscript highlights that existing data elsewhere suggest that some type of dedifferentiation mechanism (namely re-embryonalization) mediates posterior regeneration in *Platynereis*. While lines 419-421 and 515-517 provide an alternative possibility to dedifferentiation, invoking progenitor cells that are specific to a given cell lineage or germ layer, dormant or quiescent stem cells, lineage-specific progenitors etc., these should be highlighted across the entirety of the article. The inferences made during the results section are exclusively focused on a hypothesis that relies on a dedifferentiation model— authors should outline alternative hypotheses that could explain the data. Hypertranscription, cellular morphologies at a given time point during regeneration (e.g., enlarged nucleoli), and the expression of stem cell- and proliferation-related genes might be suggestive of stem cells, but they are not evidence solely for dedifferentiation. See below examples of lines in the manuscript where authors focus on a dedifferentiation hypothesis and provide no alternative hypotheses:

- Line 229: “hinting at sources of dedifferentiated PSCs”
- Lines 243-244: “consistent with multiple, parallel de-differentiation events occurring in distinct populations of wound-adjacent cells”
- Line 268: “Having found evidence for distinct pools of dedifferentiated progenitors”

- Lines 311-312: “distinct populations of mesodermal and ectodermal PSCs derive from existing cells by dedifferentiation”

We thank the reviewer for this helpful suggestion and the opportunity to clarify the aspect and likelihood of dedifferentiation as mechanism behind regeneration of the segment addition zone. We addressed this point in four ways:

1) We improved the introduction of this matter (paragraphs 6 and 7) to more explicitly explain the possible sources of cells and existing literature, including evidence from *Platynereis*. This introduction now more clearly explains that dedifferentiation is broadly assumed to be the most likely mechanism, even though the molecular trajectories had remained unclear.

2) We performed additional experiments and found new evidence strengthening the dedifferentiation hypothesis. Specifically, we have used a CytoTRACE analysis to delineate molecular changes that correlate with changes in the differentiation potential. This has led us to a marker expressed in the differentiated epidermis (*col6a6*) that we could use to validate the gradual loss of epidermal cell identity accompanying the emergence of cytological stem cell features within the same cell population (Figure 2f,g).

3) Even though we believe that these data strengthen the case for dedifferentiation, we have considered the reviewer’s comments regarding the fact that dedifferentiation is only a hypothesis. Accordingly, we re-structured the Results section to more clearly present the acquisition of stem-cell properties as a hypothesis to be tested (chapter “PSCs of ecto- and mesodermal origin exhibit shared and distinct molecular signatures”, first paragraph), followed by several findings matching the predictions made based on this hypothesis (e.g. acquisition of teloblast-like morphology, expression of GMP genes, onset of proliferation).

4) The discussion section summarizing and interpreting these results (paragraph 4) presents dedifferentiation as the most likely mechanism underlying the observed processes, while offering alternative models. We now specifically mention gut tissue as a possible exception, as recent evidence indicates the significant contribution of previously cycling cells to its regeneration.

Taken together, we now more clearly explain why we consider the model of dedifferentiation as most likely, based on published literature and our own observations (including the newly added phenotype quantification experiment), but do not exclude other options which we mention explicitly. The lines in the manuscript previously presenting dedifferentiation as the only option, which the reviewer kindly pointed out, were all changed accordingly.

2) Provide intermediate cellular states during regeneration using the transgenic lines

By means of Tol2-mediated mosaic transgenesis and regeneration assays, authors claim that regeneration of new tissues will come from cells of the same cellular identity (e.g., a mesodermal clone will regenerate cells of the mesoderm). This is true if cells of that lineage, which have inserted the transgenic construct, are undergoing differentiation— an alternative explanation would be that the transgenic construct was inserted in a genomic loci that falls under the control of a lineage-specific enhancer. In the latter possibility, the transgene is expressed whenever that enhancer is active, which could happen in uninjured animals, as well as in the blastema and later regenerated tissues, in a seemingly “restricted cellular lineage”. In order to rule out that the transgene is getting expressed because it is limited to the given cellular lineage rather than by the control of an enhancer, authors should provide figures highlighting transgenic cells that are undergoing this differentiation process. While authors show in Figure 3 the start and end point of this regeneration process, it is important show the intermediate states of those cells that are differentiating.

We thank the reviewer for making this important conceptual point. Indeed, enhancer trapping would be a nominal possibility that could explain restricted expression of the transgene even if the construct was equally distributed in all tissues, or if cells were crossing tissue boundaries.

We have now better accounted for the possibility of enhancer trapping, both in our presented experiments and in the accompanying text. Briefly, the *rps9* reporter used in our study has previously been characterized to yield strong and ubiquitous expression (Backfisch et al. PLoS One 9:e93076. doi:10.1371/journal.pone.0093076). Formally, we cannot exclude that this ubiquitous expression is occasionally superseded or perturbed by a neighboring enhancer at the site of transgene insertion. Proving this is very difficult, even if we knew the exact insertion sites of the transgene in each transgenic individual.

However, considering that we have performed additional rounds of transgenesis in the revision period and now build our arguments on a final sampling of 61 individuals (described in Supplementary Data 7), we consider it highly unlikely that such effects can explain the majority of our results:

- **Enhancer trapping is likely not a statistically efficient mechanism when transgenesis is performed with a strong ubiquitous promoter. Enhancer trapping experiments are typically conducted with minimal basal promoters. In the case of *rps9*, the transgene would need to be inserted near a strong repressor sequence for the *rps9* promoter to be perturbed. Whereas in our transgenic individuals, we observe instances where expression is variable within a clonal domain, these expressions are stochastic and not tissue-specific, as would be expected in an enhancer trapping event. This is why we interpret these stochastic expressions as classical variegation.**

- The seven classes of clonal domains that we interpret in our results are all obtained repeatedly. Given the diversity of genes and regulated patterns encoded in the target genome, one would expect a great diversity of cell or tissue-specific patterns to emerge if enhancer trapping was statistically significant. Instead, we consistently obtained each domain numerous times, suggesting that they are indeed clonal (Suppl data 7, part C)

Taken together, even the potential occurrence of enhancer trapping in a few cases would not affect the main conclusions drawn from this part of our study, namely the rigid lineage restriction of posterior stem cells and their regeneration from equally restricted lineages of cells in the trunk of *Platynereis*.

Finally, besides these statistical arguments, we also took into account the remarks of the reviewer that cells expressing the transgene in a clonal way should show a continuity of expression through divisions and differentiation.

We unfortunately still lack the tools to label individual cells in the animal and/or follow labeled cells in extended time courses that would allow us to trace individual cell divisions in development or regeneration.

However, we now include images that show clearly that the inserted constructs are able to equally express in both PSCs and differentiated cells: In Fig. 3 (g-i), we show that a specimen classified as an ectodermal integration expresses the transgene at comparable levels both in bottleneck-shaped PSCs with large nuclei/nucleoli (marked with arrowheads in h) and in transversely elongated columnar progenitor cells and squamous epidermal differentiated cells adjacent to these PSCs.

Likewise, we have managed to repeatedly image selected, labeled individuals at distinct time points during the regenerative process, and now provide high-resolution views of one specimen in the revised Fig. 4a-e. From these data, it is evident that the *rps9* construct that initially expressed in the adult epidermis (Fig. 4a), is later expressed in both PSCs (marked in Fig. 4c/d, schematized in Fig. 4e/f) and in the adjacent segment progenitor cells and differentiated cells that should have emerged from these PSCs.

3) Experiments and quantifications to strengthen authors' claims

Authors provide evidence showing that populations of cells adjacent to the wound, inferred to be the ectodermal and mesodermal PSCs, are expressing top markers of their respective clusters (cluster 0 and cluster 1, respectively). Based on the computational analysis of the single-cell RNA-seq datasets, these two major clusters show high expression of GMP-related genes and genes associated with proliferation, leading authors to claim that these cells adjacent to the wound “express stem cell-related genes and also enter the cell cycle upon injury” (lines 104-106), while also claiming that there is an “absence of obvious proliferation” (lines 518-519) during the sampled regeneration time points. Authors should clarify those statements and also provide functional data that shows that these wound-adjacent cells are expressing the

presented GMP-related genes (*piwi*, *vasa*, *nanos*), and that they are proliferative during regeneration (or alternatively show that there is no obvious proliferation response).

We thank the reviewer for raising this conceptual point, and to point out that the remarks about the absence of proliferation seemed contradictory to the idea that *bona fide* stem cells are dividing.

Concerning proliferation, as demonstrated by previous studies like Planques et al (Dev Biol 445:189–210. doi:10.1016/j.ydbio.2018.11.004), the initial 24h of regeneration essentially occur in the absence of proliferation. Proliferation initiates after this. This is now more clearly explained in the revised introduction (Line 117 and following) which should help to better contextualize our study.

Our work on the rapid expression of GMP-related genes (and the concomitant down-regulation of differentiation markers like *col6a6*) argues that there is a significant change in the molecular identity of cells preceding the proliferation phase, consistent with the idea that new stem cells emerge from the differentiated tissue that are then entering the cell cycle, as pointed out in the comment cited by the reviewer.

In keeping with the suggestions of the reviewer, we now provide new experimental data that clearly demonstrate that indeed, the GMP marker *piwi* is expressed in those predicted stem cell populations, and that these cells also incorporate EdU (Fig. 2h-k). In addition, we also show that the respective cells exhibit the enlarged nuclei and nucleoli that characterize stem cells on the cytological level.

The corresponding text has been adjusted accordingly (section “PSCs of ecto- and mesodermal origin exhibit shared and distinct molecular signatures”).

In addition, images in Figure 2 are shown to highlight the cells with enlarged nucleoli, suggestive of stem cells, in stages 1 and 3. It would be helpful to establish a definition of “enlarged nucleoli” (how to distinguish such a nucleolus from other nucleolus) and provide quantification of those cells with enlarged nucleoli across the regeneration time points, starting at stage zero, to assess whether there is a change of the number of cells with this morphology as regeneration progresses.

We thank the reviewer for encouraging us to describe this phenomenon in a more quantitative manner.

As outlined in our introduction (referring to the work of Hofmann in 1966), enlarged nucleoli and nuclei have long been considered diagnostic cytological mark for putative stem cells / “re-embryonized” cells. While there is no absolute definition of what should be considered enlarged at this early stage (and absolute values will surely also depend on the details of fixation, labeling and visualization), nuclear and nucleolar areas within and outside of the zone of putative PSCs had already been quantified at a much later

stage of regenerative growth (samples fixed 15 days after amputation; Fig. 8 of Gazave et al. Dev Biol 382:246–267. doi:10.1016/j.ydbio.2013.07.013). Those quantitative data supported statistically significantly increased areas of both nuclei and nucleoli in putative PSCs (both approximately 2fold the area of segment progenitor cells used for comparison).

In order to address the reviewer's point, we have therefore also undertaken a systematic quantification analysis, comparing both nuclear and nucleolar areas between samples taken from freshly amputated animals (0hpa), at 12hpa, and at 48hpa. For each time point, 150 nuclei / nucleoli were evaluated (50 each from 3 distinct individuals). Already at 12hpa, the nuclear areas had increased to ~1.3 fold of their comparable values at 0hpa. For nucleoli, this increase was even more pronounced, reaching almost 6-fold values (7.5fold at 48hpa). For both nuclear and nucleolar areas, the respective increases were statistically highly significant, in full support of the claims of our study. The results of the quantification are presented in the revised Fig. 2 (Fig. 2f,g), and raw data are provided in the Supplementary Data file 8.

Minor comments:

- Lines 205-207 and Figure 1e: Why is the posterior gut marker considered to be 're-emerged' at 12-72hpa? Wouldn't the 0hpa time point samples also contain foxA+ cells in the gut? Also, there is a strong expression of foxA in neuronal cells that doesn't seem to be mentioned in the text.

One of the aspects that might have been confusing in the original version of the manuscript was that it was not clearly pointed out that cell populations describing the gut were split into two clusters: one population (cluster 16) of midgut identity and a separate population (cluster 4) of hindgut identity that only emerged after injury. The most plausible explanation is that the gut in the area of the injury quickly changes molecular identity to replace the missing hindgut.

In the section describing the scRNAseq clusters we now more explicitly discuss this case as well as a similar case (cluster 14/8) for presumptive smooth muscle cells. We have also acknowledged the expression of *foxA* (and also *cdx*) in a small population of neurons that we have not attempted to characterize in more detail, but that might hint at a similar molecular difference in identities depending on the position of tissue along the antero-posterior axis.

We hope that these clarifications help to improve the understanding of this section.

- While seeing the regeneration of new tissues in Figure 3H, it is harder to tell where regeneration is happening in Figures 3I and 3J. Perhaps adding markers or arranging the panels will make it easier to identify the regeneration of these new tissues.

We thank the reviewer for suggesting these changes and agree that a rearrangement of the figure is helpful to distinguish between the growing versus regenerating tissues. In line with this suggestion, the original Figure 3 has now been split into two Figures (Fig. 3, Fig. 4). First we focus on the restriction of distinct developmental compartments in uninjured growing transgenic animals (Fig. 3 b-e) as well on the continuity of label through proliferation and differentiation of putative stem cells in the segment addition zone (Fig.3 f-i). Next, we show the persistence of the label after amputation during the regeneration process (Fig.4 a-f) and address the compartment restriction in regenerating tissue after injury in animals expressing distinct clonal labels and clearly label pre and post amputation state (Fig.4, g-l).

Reviewer #3 (Remarks to the Author):

The manuscript provided by Raible, Balavoine, Özpolat and colleagues is a formidable example for a study significantly advancing our understanding in a field of general interest such as posterior regeneration, in a non-model species that is especially relevant for this field. The regeneration of the posterior body axis occurs in vertebrates such as axolotl; yet, single-cell resolution has not yet been achieved for this process. In this study, the authors have chosen the polychaete *Platynereis dumerilii*, which has already been introduced as a model for posterior regeneration. Instead of a totipotent blastema they identify distinct stem cells for ectodermal and mesodermal lineages, forming from the epidermis and from the coelomic epithelium, respectively. Furthermore, taking advantage of the generation of transient transgenic F0 animals showing clonal inheritance of a ubiquitously expressed transgenes, they provide evidence for strict lineage restriction during regenerative growth which traces back to developmental compartments. The study thus provides a first single-cell resolution framework for an important experimental paradigm in posterior regeneration and should thus spark considerable interest of an extended readership. I have some more relevant and few minor comments that the authors should address before publication.

1) What happens to the gut? The authors find convincing upregulation of specifying transcription factors such as *cdx* and *FoxA* in clusters 11 and 20 which represent the gut but then they describe stem cells only for epidermis-derived ectodermal cells and coelomic epithelium-derived mesodermal cells. **How do endodermal mid- and hindgut cells form?**

The authors speculate that endodermal stem cells “are spread in a diffuse way along the length of the trunk”. Can this be substantiated by *cdx* and *foxA* expression?

In our data, we found strong signatures for re-patterning (morphallaxis) in gut cells, but no clear signatures of stem cell activation like we revealed in epidermal and coelomic tissues. The lack of a molecular signature for endodermal stem cells has precluded us to assess whether such cells are or are not distributed along the gut.

Overall, however, our data are in accordance with previously published evidence for a distinct mode of gut regeneration compared to tissues of ecto- and mesodermal origin, in which highly proliferating, putative progenitor cells, which can be found in gut tissues along the body axis, predominantly contribute to regeneration of this tissue (Planques et al. *Dev Biol* 2019, 445:189–210. doi:10.1016/j.ydbio.2018.11.004; Bideau et al. *Development*. 2024 Oct 15;151(20):dev202452. doi: 10.1242/dev.202452).

We clarified our findings on *foxA* (Results section), which is associated with a posteriorization process, rather than stem cells *per se*. Our findings and previous publications (Kostyuchenko et al *Dev Dyn*. 2019 Aug;248(8):728-743. doi: 10.1002/dvdy.7) indicate that, upon amputation, wound-adjacent gut tissue is broadly re-patterned towards a posterior identity, which includes the expression of *foxA*.

Given the continuity of coelomic and gut epithelium – was there ever a co-labelling of the two tissue types?

We investigated a clone restricted to gut-tissue (Figure 4i,l; Suppl. Data 7 clone N21A), and no label was found in surrounding mesodermal tissues even after repeat amputation and regeneration. As described in the manuscript, parallel insertions have also led to more complex patterns (such as labeled cells in both endo- and ectodermal tissues), but also these did not exhibit mesodermal labeling after regeneration (clones N13, N21B, N40, O09)

2) The authors introduce the concept of hypertranscriptomic stem cells recognized by high UMI count. Now, high UMI counts can also be accounted for by doublet formation. Have the authors controlled for doublets?

We thank the reviewer for suggesting this additional quality check. We have now performed a doublet analysis. Whereas this has led us to label a small cluster (cluster 26) as potential doublet (see revised Fig. 1 and accompanying text), this analysis yielded no evidence for more doublets among high UMI or CytoTRACE cells (Suppl. Figure 3 h-k; text line 222 “Algorithmic prediction...”)

3) How about additional segmented structures as chaetae and nephridia: Did the authors find evidence for new proliferative lineages also for these structures?

Due to our sampling scheme that always included the adjacent adult segment, our scRNAseq dataset indeed likely covers chaetal sac cells (cluster 24) and nephridial cells (likely represented by cluster 34). Given that both of these clusters are rather small, it would be difficult to spot any significant emergence of stem cell markers in these cell populations, unless it was a strong increase. Our analysis has at least not yielded any specific evidence for return of cells in these populations to a more stem-like character.

Should the reviewer’s question have aimed at the question whether we have found evidence for the re-emergence of these cell populations in the regenerate, we would like to clarify that the scRNAseq dataset was designed to capture the initial stages of regeneration up to the re-establishment of stem cells, but not the stage where these stem cells give rise to differentiated tissue. Hence, our scRNAseq dataset does not (yet) show a connection between the respective *bona fide* stem cells and newly formed differentiated tissue, including chaetal sac cells or presumptive nephridial cells.

4) What are the extra nuclei labelled in the endodermal clone of 3e, spreading out in a bilaterally symmetrical pattern into the pygidium?

We thank the reviewer for raising this question. The identity of background staining might have been unclear in the original version. We now include a detailed overview of

the imaging view and state clearly the expected staining background for each channel (Suppl. data 7, page3). Additionally, we now label background staining directly on the panels and state it in the legend (Fig. 3 e).

5) In their very insightful discussion the authors discuss similarities in the roles of *msx* and *sp9* in axolotl, as well as of *prrx* in chicken and mouse limb development. How does their new paradigm relate to posterior axis regeneration as occurs in salamander?

We thank the reviewer for highlighting this important aspect of our work and for suggesting additional avenues to use our data for in a comparative way. A comparison to salamander axis regeneration is challenging, as little suitable data currently exists, and gene homologies need to be established for reliable comparisons. A recent preprint (Masselink et al, bioRxiv, <https://doi.org/10.1101/2024.01.31.577464>) investigates tail regeneration in the axolotl salamander, revealing a mode of regeneration distinct to embryogenesis and development. A recent single cell study of tail regeneration in the green anole lizard, *Anolis carolinensis*, similarly found blastemal cells distinct to those used during normal development (Vonk et al, Nat Commun. 2023 Aug 10;14(1):4489. doi: 10.1038/s41467-023-40206-z).

Our data, as discussed in the manuscript, show parallels to axolotl limb regeneration, which in turn reiterates developmental processes. So while parallels between *Platynereis* tail - and salamander limb regeneration are intriguing, salamander axis regeneration might use unique, different mechanisms. Future comparative studies might still be fruitful in understanding the differences and similarities between salamander limb- and tail regeneration and *Platynereis*, but are outside the scope of this manuscript.

In summary, the authors are to be congratulated for an insightful and thought-provoking study that will significantly advance the field of axis regeneration.

We thank the reviewer for their very encouraging remarks.